# Bile acid-gut microbiota imbalance in cholestasis and its long-term effect in mice

Xin Yang,[1,2,3] Yuesong Xu,[1] Jie Li,[1] Ximing Ran,[4] Zhihao Gu,[5] Linfeng Song,[6] Lei Zhang,[1] Li Wen,[3] Guang Ji,[7] Ruirui Wang[1]

**ABSTRACT** Cholestasis is a common morbid state that may occur in different phases; however, a comprehensive evaluation of the long-term effect post-recovery is still lacking. In the hepatic cholestasis mouse model, which was induced by a temporary complete blockage of the bile duct, the stasis of bile acids and liver damage typically recovered within a short period. However, we found that the temporary hepatic cholestasis had a long-term effect on gut microbiota dysbiosis, including overgrowth of small intestinal bacteria, decreased diversity of the gut microbiota, and an overall imbalance in its composition accompanied by an elevated inflammation level. Additionally, we observed an increase in *Escherichia-Shigella* (represented by ASV136078), rich in virulence factors, in both small and large intestines following cholestasis. To confirm the causal role of dysregulated gut microbiota in promoting hepatic inflammation and injury, we conducted gut microbiota transplantation into germ-free mice. We found that recipient mice transplanted with feces from cholestasis mice exhibited liver inflammation, damage, and accumulation of hepatic bile acids. In conclusion, our study demonstrates that cholestasis disrupts the overall load and structural composition of the gut microbiota in mice, and these adverse effects persist after recovery from cholestatic liver injury. This finding suggests the importance of monitoring the structural composition of the gut microbiota in patients with cholestasis and during their recovery.

**IMPORTANCE** Our pre-clinical study using a mouse model of cholestasis underscores that cholestasis not only disrupts the equilibrium and structural configuration of the gut microbiota but also emphasizes the persistence of these adverse effects even after bile stasis restoration. This suggests the need of monitoring and initiating interventions for gut microbiota structural restoration in patients with cholestasis during and after recovery. We believe that our study contributes to novel and better understanding of the intricate interplay among bile acid homeostasis, gut microbiota, and cholestasis-associated complications. Our pre-clinical findings may provide implications for the clinical management of patients with cholestasis.

**KEYWORDS** chemically induced cholestasis, gut pathobionts, *Escherichia-Shigella*, small intestinal bacterial overgrowth, intestinal bacterial translocation

Cholestasis, characterized by the inability of bile to expel from the liver normally or impairment of bile formation, is a common clinical condition that initiates or complicates chronic liver diseases (1, 2). The etiology of cholestasis can be both extrahepatic and intrahepatic pathogenic causes including genetic susceptibility, drugs, biliary obstruction, and inflammation (1). An increased frequency of cholestasis was also observed during pregnancy and in neonatal period (3).

Cholestasis is usually benign and self-limiting; however, the retention of bile in the liver and systemic circulation can result in numerous biochemical abnormalities (4), especially during pregnancy as it will affect both pregnant women and their fetuses. For

**Ad Hoc Peer Reviewers** Anand Kumar, Los Alamos National Laboratory Bioscience Division, Los Alamos, New Mexico, USA; Sudip Das, Universitat Bern, Bern, Switzerland; Kazuyuki Kasahara, Nanyang Technological University, Singapore, Singapore

Address correspondence to Li Wen, li.wen@yale.edu, Guang Ji, jg@shutcm.edu.cn, or Ruirui Wang, wangruirui@shutcm.edu.cn.

The authors declare no conflict of interest.

See the funding table on p. 16.

example, approximately 6% of neonates with cholestasis develop cirrhosis after recovery (5). It has also been reported that intrahepatic cholestasis of pregnancy (ICP) could increase the adverse risk to fetuses (6), including neonatal depression, preterm birth, and even stillbirth (7–9). Although current medical interventions are generally effective in facilitating recovery in most patients with cholestasis, there remains a paucity of mechanistic understanding regarding the comprehensive and derivative assessment of cholestasis.

The amphipathic nature of bile acids enables cytotoxic effects (10, 11), which act as important environmental selection pressures to maintain homeostasis in the number and species of the intestinal microbiota (11, 12). An imbalance in the gut microbiota, partially driven by bile acid metabolism, may play a role in the development of various diseases in the host (12–16). In addition to the structure of the gut microbiota, the load of gut bacteria may be regulated by bile acids. It has been reported that small intestinal bacterial overgrowth (SIBO) occurs in patients with progressive familial intrahepatic cholestasis in hydrogen–methane breath test (17).

Although the dysbiosis of microbiota in the colon during cholestasis has been well demonstrated (2, 18), there are still many unknowns, especially the systematic changes of the inherent structural dysbiosis as well as the longitude effects of microbiota dysbiosis post cholestasis recovery. In this study, we analyzed bile acid concentrations in the liver, fecal samples, and serum, along with the composition and abundance of gut microbiota in α-Naphthyl-isothiocyanate (ANIT)-induced cholestasis. We discovered the enrichment of *Escherichia-Shigella* strain (ASV136078) that is associated with serum bile acids. Furthermore, we identified virulence factors in the dysbiosis of the gut microbiota by shotgun metagenomic sequencing. We further evaluated the role of cholestasis-induced gut microbiota in liver inflammation and disrupted bile acid metabolism in germ-free mice. We found that recipient mice transplanted with feces from a cholestasis mice exhibited liver inflammation, damage, and accumulation of hepatic bile acids. These findings suggest that dysbiosis of the gut microbiota could persist and affect enterohepatic circulation even after cholestasis recovery in mice.

## MATERIALS AND METHODS

### Animal experiments

#### Mice

Eight-week-old SPF male mice (C57BL/6) were obtained and maintained in an SPF environment at the animal facility of the Shanghai Research Center of Southern Model Organisms (Shanghai, China). Mice (4–5 mice/cage) were housed in Individually Ventilated Cage (IVC) system (dimensions 325 × 210 × 180 mm; the Suzhou Fengshi Laboratory Animal Equipment Co., Ltd). The mice were fed with an irradiated sterilized normal chow diet and were housed in a room maintained at 22.0 ± 1℃ with 30%–70% humidity and a 12 h light/dark phase cycle (lights on at 07:00 am and off at 07:00 pm).

Eight-week-old germ-free male C57BL/6 mice were maintained in flexible-film plastic isolators at the Laboratory Animal Center of SLAC, Inc. (Shanghai, China). The SPF mice and germ-free mice were fed an autoclaved normal chow diet (10% energy from fat; 3.25 kcal/g; P1103F-50, SLAC). The mice were maintained under the same feeding conditions as those in the previous experiment. All animal experiments were approved by the Institutional Animal Care and Use Committee (IACUC) of the Shanghai University of Traditional Chinese Medicine (no. SMOC-IACUC 2018-0036).

#### Experiments

##### Short-term experiment

After 1-week of acclimation, the mice were randomly assigned to two groups: (i) normal control (NC) group (*n* = 12), mice had drinking water for 8 days; (ii) ANIT group (*n* = 12),

mice were orally gavaged with 75 mg/kg ANIT (Sigma-Aldrich, St. Louis, MO, USA) once. The experiments were repeated once, separately.

### Long-term experiment

After 1-week of acclimation, the mice were randomly assigned to two groups: (i) NC group ($n = 12$), control mice had drinking water for 28 days; (ii) ANIT group ($n = 13$), mice were administered with 75 mg/kg ANIT (Sigma-Aldrich) through oral gavage for 28 days.

### Fecal microbiome transfer experiment

The germ-free mice were randomly assigned to two groups, which were kept in separate gnotobiotic isolator. (i) $GF_{NC}$ group ($n = 5$), germ-free mice inoculated with fecal samples NC mice on day 2 of the short-term experiment. (ii) $GF_{ANIT}$ group ($n = 5$), germ-free mice inoculated with fecal samples from the ANIT group on day 2 after ANIT treatment from the short-term experiment. Seven days after the fecal microbiota transplantation was performed, the mice were euthanized, and the tissues of interest were harvested.

## Preparation of inoculum from mice feces

Fresh fecal samples from each mouse in the NC and ANIT groups (on the second day of the short-term experiment) were collected. The pooled fecal samples in the same group were mixed in anaerobic sterile Ringer working buffer (9 g/L of sodium chloride, 0.4 g/L of potassium chloride, 0.25 g/L of calcium chloride dehydrate and 0.05% (w/v) L-cysteine hydrochloride) in an anaerobic chamber (80% $N_2$:10% $CO_2$:10% $H_2$) (19), and diluted to 50 times. The suspensions were then added to 20% (w/v) skim milk (LP0031, Oxford, UK), and the final fecal suspensions were diluted 100 times. The fecal suspensions were preserved at $-80°C$ until inoculation.

## alanine aminotransferase (ALT) and aspartate aminotransferase (AST) measurement

Serum ALT and AST were quantified using colorimetric kits (C009-2-1 and C010-2-1, respectively) from the Nanjing Jiancheng Bioengineering Institute (Nanjing, China). A standard curve was prepared by serially diluting a known concentration of recombinant ALT (or recombinant AST) standard to eight different concentration gradients.

## Detection of total bile acid in fecal and ileal contents

The bile acid contents in fecal and ileal samples were quantified using a colorimetric kit (E003-2-1) from Nanjing Jiancheng Bioengineering Institute (Nanjing, China).

## Preparation of liver homogenate and total bile acid detection

Frozen liver samples were homogenized in a homogenizing buffer (pH 7.4, 0.01 mol/L Tris-HCl, 0.1 mmol/L EDTA-2Na, 0.8% NaCl) at the liver weight vs buffer volume of 1:9), followed by centrifugation (2,000 g for 25 min at 4°C). The supernatants were diluted 20 times with a homogenizing buffer (pH 7.4, 0.01 mol/L Tris-HCl, 0.1 mmol/L EDTA-2Na, 0.8% NaCl) and used to measure concentrations of hepatic total bile acid using the kits described above.

## mRNA expression of genes related with gut barrier integrity

Total RNA from the colon, ileum, and liver was extracted using an RNeasy Plus Universal Tissue Mini Kit (73404; Qiagen, Duesseldorf, Germany). RNA was reverse-transcribed into cDNA using SuperScript III First-Strand Synthesis (18080051, Invitrogen, Carlsbad, CA, USA). RT-qPCR was performed with a Light Cycler 96 (Roche, Geneva, Switzerland) using iQ SYBR Green Supermix (170-8882AP, BIO-RAD, CA, USA). The PCR conditions were 95°C for 3 min, followed by 40 cycles of 95°C for 20 s, 56°C for 30 s, and 72°C for 30 s, and plate

reads for 5 s at 80°C. Gene expression levels were determined using the comparative $\Delta\Delta C_T$ method ($2^{-\Delta\Delta C_T}$ method), with the β-actin gene serving as the reference gene. The forward (F) and reverse (R) primer sequences are listed in Table S1.

## Histopathology

Fresh liver tissues were fixed with 4% paraformaldehyde for 48 h, followed by embedding in paraffin and sectioning (4 µm). The sectioned slides were stained with hematoxylin and eosin (H&E) (G1003; Wuhan Servicebio technology Ltd., Wuhan, China). Digital images of sections were acquired using a Leica DMRBE microscope.

## Immunohistochemistry staining of TNFα and IL-6 in the liver

After deparaffinization and antigen retrieval, blank colon sections were incubated overnight at room temperature with anti-mouse TNFα (Servicebio, GB11188-100, 1:200) and IL-6 (Servicebio, GB11117-100, 1:200) antibodies separately. The sections were incubated with an Horseradish Peroxidase (HRP)-conjugated Goat Anti-Mouse IgG (H + L) or HRP-conjugated Goat Anti-Rabbit IgG (H + L) secondary antibody (Servicebio, GB23303 and GB23301, 1:200) for 50 min at room temperature. After washing with PBS, DAB EnVision kit (catalog no. K5007; Dako, Copenhagen, Denmark) was used to develop the color; TNFα and IL-6 were stained brown, while nuclei were blue. Images of each colon section were obtained in tripartite by experienced staff members who were blind to the experiment under ×200 magnification using a Leica DMRBE microscope and were analyzed using Image Pro Plus 6.0, as previously described (20). Integrated optical density (IOD) values were transformed in log10.

## Assessment of blood bile acid metabolome

Comprehensive profiling and quantitation of bacterial metabolites were performed (Metabo-Profile Inc., Shanghai, China) using previously published methods (21, 22).

## Bacterial load in the ileal contents and fecal samples

Bacterial load was quantified by qPCR with a 466 bp length sequence located in the 16S rRNA gene using the primers Uni331F (TCCTACGGGAGGCAGCAGT) and Uni797R (GGACTACCAGGGTATCTAATCCTGTT) (23). qPCR was performed as described earlier (24). A full-length 16S rRNA gene of an *Akkermansia muciniphila* strain derived from the mouse intestine was used to plot a standard curve to calculate the copies of 16S rRNA genes in the samples.

## Gut microbiota profiling

### High-throughput 16S rRNA gene sequencing

Total genomic DNA was extracted from fecal samples using the E.Z.N.A. Soil DNA Kit (Omega Bio-tek, Norcross, GA, U.S.) according to the manufacturer's instructions. The concentration and purity of the extracted DNA were determined using Qubit4.0 and NanoDrop2000, respectively. The quality of the DNA extract was validated on a 1% agarose gel.

A sequencing library of the V3-V4 region of the 16S rRNA gene was constructed according to the manufacturer's instructions (Part # 15044223Rev. B; Illumina Inc., San Diego, CA, USA), as previously described (25) and sequenced on the Illumina MiSeq platform (Illumina Inc.). The sequencing service was provided by the Personal Biotechnology Co., Ltd. Shanghai, China. Raw paired-end reads were processed and analyzed using QIIME2. Demultiplexed sequence data were imported into QIIME2; the adapters and primers were trimmed. The amplicon sequence variants (ASVs) from each sample were inferred using the DADA2 pipeline for filtering, dereplication, sample inference, merging of paired-end reads, and chimera identification. In the process of running the DADA2 pipeline, based on the quality profile of the data, forward and reverse reads were

trimmed to ensure that the median quality score for each position was above 32. The taxonomy of all ASVs was annotated using the SILVA (v132) reference database (26). All the samples were rarefied to 10,000 reads per sample for downstream analyses. Principal coordinate analysis (PCoA) of ASVs based on the Bray-Curtis distance was performed using the QIIME2. Statistical significance was assessed using permutational multivariate analysis of variance (PERMANOVA) with 9,999 permutations, and *P* values were adjusted for multiple comparisons with the Benjamini–Hochberg method (27).

Random Forest and Linear Discriminant Analysis Effect Size (LEfSe) models were established to identify specific ASVs that contributed to the segregation of the gut microbial structure in NC and ANIT mice. Sequence data associated with this project were deposited in the NCBI database. Short Read Archive database (Project ID: PRJNA974219, Study ID: SRP438742).

## Metagenomic sequencing

The DNA extract was fragmented to an average size of approximately 400 bp using a Covaris M220 for paired-end library construction. A paired-end library was constructed using NEXTFLEX Rapid DNA-Seq (Bioo Scientific, Austin, TX, USA). Adapters containing the full complement of the sequencing primer hybridization sites were ligated to the blunt ends of the fragments. Paired-end sequencing was performed on an Illumina NovaSeq (Illumina Inc.) at Honsunbio Technology Co., Ltd. (Shanghai, China) using NovaSeq Reagent Kits, according to the manufacturer's instructions (www.illumina.com). Sequence data associated with this project were deposited in the NCBI database. Short Read Archive database (Project ID: PRJNA975699, Study ID: SRP439239).

## Sequence quality control and genome assembly

The paired-end Illumina reads were trimmed of adaptors and low-quality reads (length < 50 bp, with a quality value < 20, or having N bases) were removed using fastp[1] (https://github.com/OpenGene/fastp, version 0.20.0).

Reads were aligned to the human genome using Burrows-Wheeler Aligner (BWA) (28) (http://bio-bwa.sourceforge.net, version 0.7.9a), and hits associated with the reads and their mated reads were removed.

Metagenomics data were assembled using MEGAHIT (29) (https://github.com/voutcn/megahit, version 1.1.2), which uses succinct de Bruijn graphs. Contigs with a length of or over 300 bp were selected as the final assembly results and used for further gene prediction and annotation.

## Metagenomic binning

Contigs with a length ≥ 1,000 bp were selected as the final assembling result, and then the contigs were used for further binning to retrieve Genome bins (or metagenome-assembled genomes, MAGs). Binning was performed using MetaBAT (version 2.12.1). The contamination of each bin was assessed by examining the GC content, coverage, and tetranucleotide frequency (TNF) of the contigs within each bin. Contigs with better quality were selected based on the assessment results, and contaminated sequences were manually removed to correct the bins.

The completeness, contamination, and strain heterogeneity of the bins were estimated by CheckM (version 1.0.11), and the bins with more than 50% completeness and <10% contamination were kept for downstream analysis. Bowtie2 (version 2.2.9) was used for short-read alignment. Taking into account the alignment results between the clean reads and the contigs in each bin, all the mapped reads in the sample were merged and de-duplicated. Afterwards, these reads were re-assembled using SPAdes (version 3.12.0).

Species-level annotation and classification of the obtained bins were performed using the Genome Taxonomy Database Toolkit (GTDB-Tk) (GTDB-Tk v0.3.2) (30).

## De novo non-redundant metagenomic gene catalog construction and gene abundance profile calculations

A non-redundant gene catalog was constructed using Cluster Database at High Identity with Tolerance (CD-HIT) (31) (http://www.bioinformatics.org/cd-hit/, version 4.6.1), with 90% sequence identity and 90% coverage. Reads after quality control were mapped to the non-redundant gene catalog with 95% identity using SOAPaligner (32) (http://soap.genomics.org.cn/, version 2.21), and the gene abundance in each sample was evaluated.

### Functional annotation

Open reading frames (ORFs) from each assembled contig were predicted using MetaGene (33) (http://metagene.cb.k.u-tokyo.ac.jp/). The predicted ORFs with a length of 100 bp were retrieved and translated into amino acid sequences using the NCBI translation table (http://www.ncbi.nlm.nih.gov/Taxonomy/taxonomy-home.html/index.cgi?chapter=tgencodes#SG1).

Representative sequences of non-redundant gene catalog were aligned to the NCBI NR database with an e-value cut-off of $1e^{-5}$ using Diamond (34) (http://www.diamond-search.org/index.php, version 0.8.35) for taxonomic annotations. Clusters of orthologous groups of proteins (COG) annotation for the representative sequences were performed using Diamond (http://www.diamondsearch.org/index.php, version 0.8.35) against the eggNOG database with an e-value cutoff of $1e^{-5}$. The Kyoto Encyclopedia of Genes and Genomes (KEGG) annotation was conducted using Diamond (34) (http://www.diamondsearch.org/index.php, version 0.8.35) against the KEGG database (http://www.genome.jp/keeg/) with an e-value cutoff of $1e^{-5}$.

Carbohydrate-active enzyme annotation was conducted using hmmscan (http://hmmer.janelia.org/search/hmmscan) against the CAZy database (http://www.cazy.org/), with an e-value cutoff of $1e^{-5}$. Virulence factor annotation was conducted using Diamond (34) (http://www.diamondsearch.org/index.php, version 0.8.35) against the virulence factor database (VFDB) database (http://www.mgc.ac.cn/VFs/), with an e-value cutoff of $1e^{-5}$.

### Statistical analysis

The horizontal lines in the figures presented as point plots and/or bar plots indicate mean ± SEM, and statistical significance was assessed by two-tailed Student's $t$-test (two groups) or one-way ANOVA (>2 groups) using GraphPad Prism (version 9), with $P$ values of <0.05.

## RESULTS

### Comprehensive assessment of cholestasis-induced acute liver injury and bile acid redistribution in ANIT-treated mouse model

To explore the interaction of bile acid and gut microbiota in the recovery of cholestasis, we adopted a commonly used cholestasis mouse model induced by a biliary epithelial toxicant, ANIT, which induces short-term hepatic cholestasis (35, 36). We assessed the phenotypes of the C57BL/6 mice (6-week-old, male) on the second and eighth days after ANIT treatment by oral gavage of 75 mg/kg once. The control group comprised the mice without ANIT intake (Fig. 1A). Compared with control mice, the mice that received ANIT treatment showed a rapid loss of ~17% of body weight on day 4 but mostly recovered on day 8 (Fig. 1B). However, the liver weight percentage remained much higher in the ANIT treated mice than that of the control mice on day 8 (Fig. 1C). In line with the body weight loss, the food and water intake of the mice decreased during the first three days after ANIT treatment and fully recovered (Fig. S1A and B). Serum ALT and AST levels, which are indicative of liver damage, were significantly higher in the ANIT-treated mice than those in the untreated mice on both 2 and 8 days, and the levels of ALT

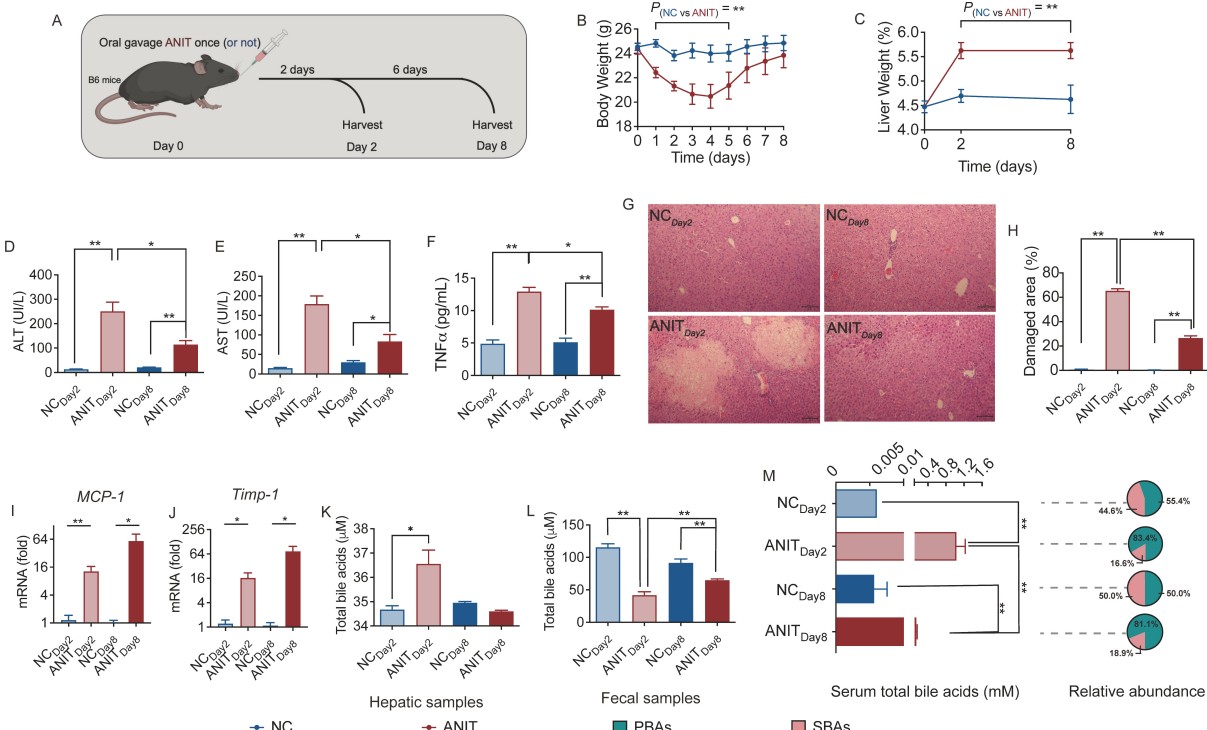

**FIG 1** Metabolic phenotypes of ANIT-induced cholestatic liver injury in mice. (A) Experimental flow diagram. (B) Body weight and (C) Liver weight, NC group, $n = 10$; ANIT group, $n = 10$. (D) Serum ALT, (E) AST and (F) TNFα concentration, NC group, $n = 10$; ANIT group, $n = 10$. (G) H&E-stained histological sections of hepatic tissue (400×, scale bar = 0.05 mm) from NC and ANIT mice, NC group, $n = 5$; ANIT group, $n = 5$. (H) Damaged area was calculated based on H&E-stained histological sections of hepatic tissue from NC and ANIT mice by Image Pro Plus software. NC group, $n = 5$; ANIT group, $n = 5$. Levels of mRNA expression of (I) MCP-1 and (J) Timp-1 gene in the hepatic tissue, NC group, $n = 6$; ANIT group, $n = 6$. Total bile acids in the (K) hepatic tissues, (L) the fecal samples and (M) the serum samples, NC group, $n = 6–10$; ANIT group, $n = 6–10$. $NC_{Day2}$ and $NC_{Day8}$ refer to day 2 and day 8 of the control mice after water treatment, respectively; $ANIT_{Day2}$ and $ANIT_{Day8}$ refer to the mice on day 2 and day 8 after 75 mg/kg ANIT gavage, respectively. PBAs = primary bile acids, SBAs = secondary bile acids. The data in (B–F) and (H–M) are expressed as mean ± SEM, and Student's $t$-test (two-tailed) was used to analyze differences between the following pairs of groups: $NC_{Day2}$ vs. $ANIT_{Day2}$, or $NC_{Day8}$ vs. $ANIT_{Day8}$. One-way ANOVA was used to analyze differences among the following four groups: $NC_{Day2}$, $ANIT_{Day2}$, $NC_{Day8}$, and $ANIT_{Day8}$. *$P < 0.05$ and **$P < 0.01$.

and AST were the highest on day 2 after ANIT treatment (Fig. 1D and E). The TNFα concentration in the serum of mice in the ANIT group significantly increased on the 2nd (highest) and 8th days, indicating systemic inflammation in the mice. The serum TNFα level on the 8th day after ANIT administration was declined compared to that on the 2nd day after ANIT administration, which suggests that the acute inflammation was gradually resolving (Fig. 1F). These mice ($ANIT_{Day2}$ and $ANIT_{Day8}$) also had liver damage, as shown by histological characteristics of necrosis and fibrosis in the liver (Fig. 1G and H). Furthermore, the hepatic expression of *MCP-1* and *Timp-1*, which are related to injury and fibrosis, was significantly upregulated in the ANIT treated mice, both $ANIT_{Day2}$ and $ANIT_{Day8}$ (Fig. 1I and J). These results indicate that one administration of ANIT resulted in cholestasis-induced liver injury for at least 8 days in mice. To verify cholestasis, we assessed total bile acids in the liver and found an obvious hepatic cholestasis on day 2, but it was fully recovered on day 8 (Fig. 1K). We also assessed fecal bile acids and found significantly lower levels of fecal bile acids on both days in the ANIT group than in the controls (Fig. 1L). The concentration of serum total bile acids increased in ANIT mice. Also, the changes of different secondary bile acids varied, with some increased and some decreased (Fig. S4B), but the ratio of secondary bile acids decreased in ANIT mice (Fig. 1M). These results indicated that bile acids-related metabolic disturbance and liver damage were still present at the recovery time (day 8) in the ANIT treated mice. Next, we detected the gene expression levels of bile acid transport-related proteins in the liver

and ileum, and the results showed that the gene expression levels of these functional proteins in the liver and ileum of ANIT-treated mice were significantly different from those in the non-ANIT-treated group, especially on day 2 and some alterations remained till day 8 (Fig. S2A and B). Thus, our results demonstrated that oral gavage of 75 mg/kg ANIT induces short-term hepatic cholestasis in mice. However, a more comprehensive investigation is needed to understand the systemic implications of cholestasis.

## Acute cholestasis enriches *Escherichia-Shigella* and increases bacterial load in the intestinal tract of the mice

Bile acid in the gut, as an important environmental stress factor, significantly affects the composition and stable abundance of the gut microbiota (37–39). To investigate the interaction between gut microbiota and bile acid during the early acute phase and later recovery phase of cholestasis, we performed high-throughput sequencing of the V3-V4 region of the 16S rRNA gene using fecal samples at different time points post ANIT treatment. We found that the gut microbiota structure in the ANIT-treated mice was significantly shifted away from the controls, as shown by its reduced alpha diversity (Fig. 2A and C; Fig. S3A through G) and richness (Fig. 2B). There were also significant differences in the PCoA score plot of Bray-Curtis distance and Weighted UniFrac distance based on the ASVs of the 16S rRNA gene V3-V4 region ($P = 0.001$ and $P = 0.0004$ by permutational multivariate analysis of variance (PERMANOVA) test with 999 permutations, Fig. 2D and F) [Based on the microbiota that did not change over time, the microbiota structure of the control group at different time points was merged (Fig. S3H)]. Furthermore, we quantified the distance between the gut microbiota structure at different time points after ANIT treatment and those of the control group and found that the gut microbiota structure on day 2 after ANIT treatment was farthest from those of the control group. Although, the distance between the gut microbiota structure in ANIT treatment and NC groups was smaller over time, there was still significant difference between the two groups on day 8, the last time point studied (Fig. 2E and G). Interestingly, these dynamic changes were accompanied with the clinical and patho-histological features of cholestasis and liver injury (Fig. 1). Thus, ANIT-induced cholestasis in mice is accompanied by significant changes in the gut microbiota.

To identify key members of the gut microbiota enriched by ANIT, we first analyzed ASVs in the gut microbiota that showed significant differences between ANIT-treated and control mice using a random forest model (Fig. 2H). Notable changes were observed in the genus *Escherichia-Shigella* (Fig. 2I), which is a major opportunistic gram-negative pathogen. Specifically, four ASVs (ASV136078, ASV11534, ASV61708, and ASV159897) of this genus were enriched after ANIT treatment (Fig. 2H). To probe the link of bile acids with *Escherichia-Shigella*, we performed Spearman correlation analysis on the matrix of 45 bile acid concentrations in blood detected by Ultra Performance Liquid Chromatography-Mass Spectrometry/Mass Spectrometry (UPLC/MS/MS) and the relative abundance of ASVs, and we found that ASV136078 (*Escherichia-Shigella*) was significantly positively correlated with 16 bile acids, most of which were secondary bile acids (Fig. 2J, top row). In addition, 10 ASVs belonging to the Muribaculaceae family were negatively correlated with some serum bile acids (Fig. 2J, row 11–17, 20, 22–23). Additionally, the quantitation result of total bacteria in fecal samples revealed that there were higher bacterial loads in fecal samples from ANIT-treated mice on day 2 and day 8 after treatment (Fig. 2K).

Furthermore, we investigated the possibility of small-intestinal bacterial growth in ANIT-treated mice (40, 41) by quantifying bacteria and high-throughput sequencing using small-intestinal content. ANIT-treated mice showed a marked increase of bacterial load in small intestine on day 8 (Fig. 2L), indicating the occurrence of SIBO. The gut microbiota structure in the ANIT-treated mice was significantly shifted away from that of the control, as shown by its increased alpha diversity (Fig. 2M; Fig. S5A and B), increased richness (Fig. 2N), and significant difference in the PCoA score plot of Bray-Curtis distance ($P = 0.001$ by PERMANOVA test with 999 permutations, Fig. S6A). Similar to that in the colon, cholestasis induced by ANIT can also increase the proportion

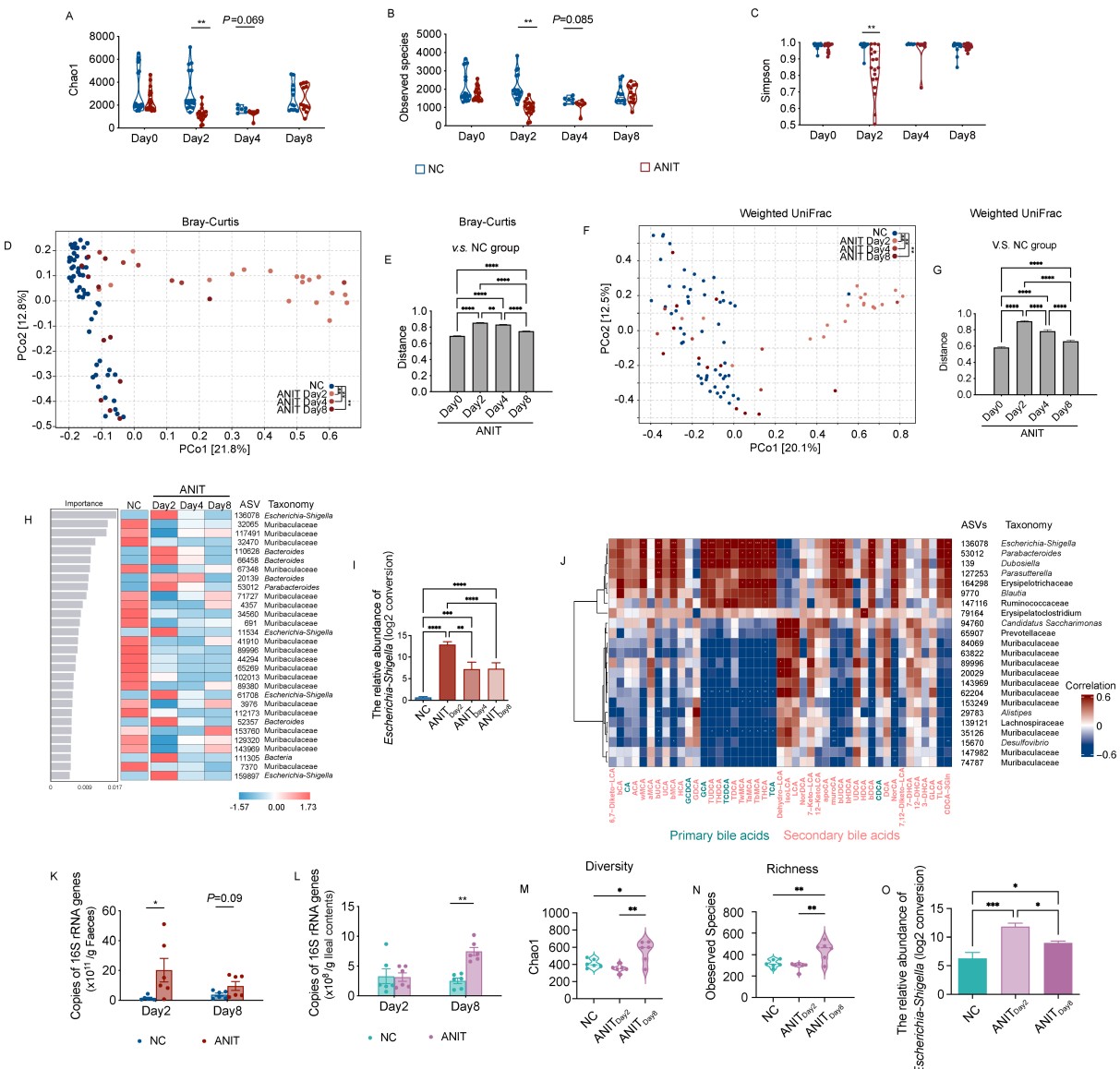

**FIG 2** ANIT-induced cholestasis alters the structure and load of the gut microbiota in mice. (A) The Chao1, (B) richness (observed species) and (C) Simpson index of gut microbiota on day 0, 2, 4, and 8 fecal samples from NC and ANIT groups. (D) Overall gut microbial structure in NC and ANIT mice presented by PCoA based on Bray-Curtis distance and (F) Weighted UniFrac distance at the ASV level. (E and G) The distance between each time points of ANIT group and NC group. For (A–G), $n$ = 6–17/group. (H) Thirty ASVs that were significantly altered after ANIT treatment, as identified using random forest models. The heat map shows the relative abundance ($\log_{10}$ transformed) of each ASV in a pooled and barcoded sample from each group of mice, NC group, $n$ = 12; ANIT group, $n$ = 16–17. (I) The relative abundance of *Escherichia-Shigella* in the fecal sample of each group, NC group, $n$ = 12; ANIT group, $n$ = 6–17. (J) Twenty-three ASVs significantly associated with bile acids in serum. The cluster tree on the left shows associations between these ASVs, as determined by the Spearman correlation coefficient based on their relative abundances among all the samples. The heat map shows the association between ASVs and bile acids. The colors denote the correlation coefficients. $P$ values were adjusted by Benjamini–Hochberg procedure (*$P$ < 0.05, **$P$ < 0.01 and ***$P$ < 0.001). The bacterial load in (K) the fecal samples and (L) the ileal content of NC and ANIT mice as measured by real-time qPCR, $n$ = 6/group. (M) The alpha diversity and (N) richness (observed species) of gut microbiota in ileal content of NC and ANIT group mice on day 2 and day 8, $n$ = 6/group. (O) The relative abundance of *Escherichia-Shigella* in the ileal content of each group, NC group, $n$ = 12; ANIT group, $n$ = 6–17. NC$_{Day2}$, on day 2 of the control mice; NC$_{Day4}$, on day 4 of the control mice; NC$_{Day8}$, on day 8 of the control mice; ANIT$_{Day2}$, on day 2 of the mice after 75 mg/kg ANIT gavage; ANIT$_{Day4}$, on day 4 of the mice after 75 mg/kg ANIT gavage; ANIT$_{Day8}$, on day 8 of the mice after 75 mg/kg ANIT gavage. The data in (A–C), (E), (G), (I), and (K–O) are expressed as mean ± SEM, Student's $t$-test (two-tailed) was used to analyze differences between the following group pairs: NC$_{Day0}$ vs. ANIT$_{Day0}$; NC$_{Day2}$ vs. ANIT$_{Day2}$; NC$_{Day4}$ vs. ANIT$_{Day4}$; and NC$_{Day8}$ vs. ANIT$_{Day8}$. One-way ANOVA was used to analyze differences when there are more than two groups. *$P$ < 0.05 and **$P$ < 0.01.

of *Escherichia-Shigella* (ASV 136078) in the ileum (Fig. 2O; Fig. S6B). Our results demonstrated that ANIT-induced cholestasis significantly altered the relative composition and absolute abundance of the small intestinal and colonic microbiota structures in mice. Moreover, the cholestasis-enriched bacteria belonging to the genus *Escherichia-Shigella* were significantly associated with serum concentrations of 16 bile acids. Thus, cholestasis affects both the composition and the bacterial load of the gut microbiota.

## Cholestasis-induced dysbiosis of gut microbiota had legacy effects

It is known that ANIT-induced cholestasis and liver injury in mice are self-limiting and quickly recovered from the cholestatic liver injury, while our results showed that various pathological phenotypes were still present on day 8 although the mice had normal body weight and hepatic bile acids. To better understand the legacy effect of transient cholestasis, we conducted a long-term ANIT-induced cholestasis experiment over 28 days. Hepatic cholestasis remained to be normal on day 14 and day 28, and liver injury was also completely recovered from day 14 and remained to be normal on day 28 (Fig. 3A through C). However, we observed different abnormalities in non-hepatic organs and tissues. We found that the spleen of the ANIT-treated mice was significantly enlarged compared with that of the control group on day 14 but became normal on day 28 (Fig. 3D). Whereas the concentrations of circulating TNFα in the ANIT-treated mice was persistently higher compared to their respective control groups on both day 14 and day 28 (Fig. 3E). Interestingly, although hepatic cholestasis was sufficiently relieved in the ANIT-treated group from day 8 (Fig. 1J), the bacterial loads in fecal and ileal samples remained significantly higher in the ANIT-treated group than in the control group on day 14 and/or day 28 (Fig. 3F and G). Moreover, the gut microbiota structure in the ANIT$_{Day28}$ groups was significantly shifted away from their controls, as shown by the reduced alpha diversity (Fig. 3H), reduced richness (Fig. 3I), and significant difference, on both day 14 and day 28, in the PCoA score plot of Bray-Curtis distance based on the ASVs of the 16S rRNA gene V3-V4 region ($P = 0.001$ by PERMANOVA test with 999 permutations, Fig. 3J). Similar to the short-term observations earlier, the accumulation of the genus *Escherichia-Shigella* represented by ASV136078 in the colon remained to be elevated (Fig. 3K and L). Thus, ANIT did not induce permanent cholestasis in mice, but the systemic inflammation and disturbed gut microbiota had not recovered after the recovery of cholestasis, indicating a legacy effect of short-term cholestasis.

## Cholestasis enriches encoding genes of bacterial virulence factors

To better understand the function of the disturbed gut microbiota in our cholestasis mouse model, we performed shotgun metagenomic sequencing using fecal samples from the NC$_{Day2}$ ($n = 6$) and ANIT$_{Day2}$ ($n = 6$) groups. Supporting our 16S rRNA sequencing data, the shotgun metagenomic sequencing results showed an increased relative abundance of the genus *Escherichia-Shigella* in the ANIT-treated group, which led us to investigating the genes for virulence factors. We annotated the functional virulence factors of the obtained genes using the VFDB (42). A total of 106 annotated virulence factor genes were significantly enriched in the ANIT-treated group, whereas none of the virulence factor genes were enriched in the control group (Fig. 4A; Table S2). We focused on ASV136078, a representative strain of *Escherichia-Shigella*, because of the persistent increase in relative abundance at the peak of cholestasis induced by ANIT (Fig. 2H through J, O, 3K and L). We constructed a phylogenetic tree using high-quality MAGs capable of extracting 16S rRNA gene sequences, with the ASV136078 sequence together. We discovered that bin 255 has the closest evolutionary relationship with ASV136078 (Fig. 4B). Further, through sequence alignment, we found that the similarity between the 16S rRNA gene sequences of bin 255 and ASV136078 is 100% (Fig. 4C; Supplement file 3). We assembled a 2.8M draft genome belonging to bin 255, the genus *Escherichia-Shigella* (Fig. 4D; Supplement file 4). We annotated the potential pathogenicity of the spliced genome drafts through the Pathogen Host Interactions (PHIs) database and the VFDB, respectively (42, 43). After setting the E-value standard to be less than 1e-5, we found

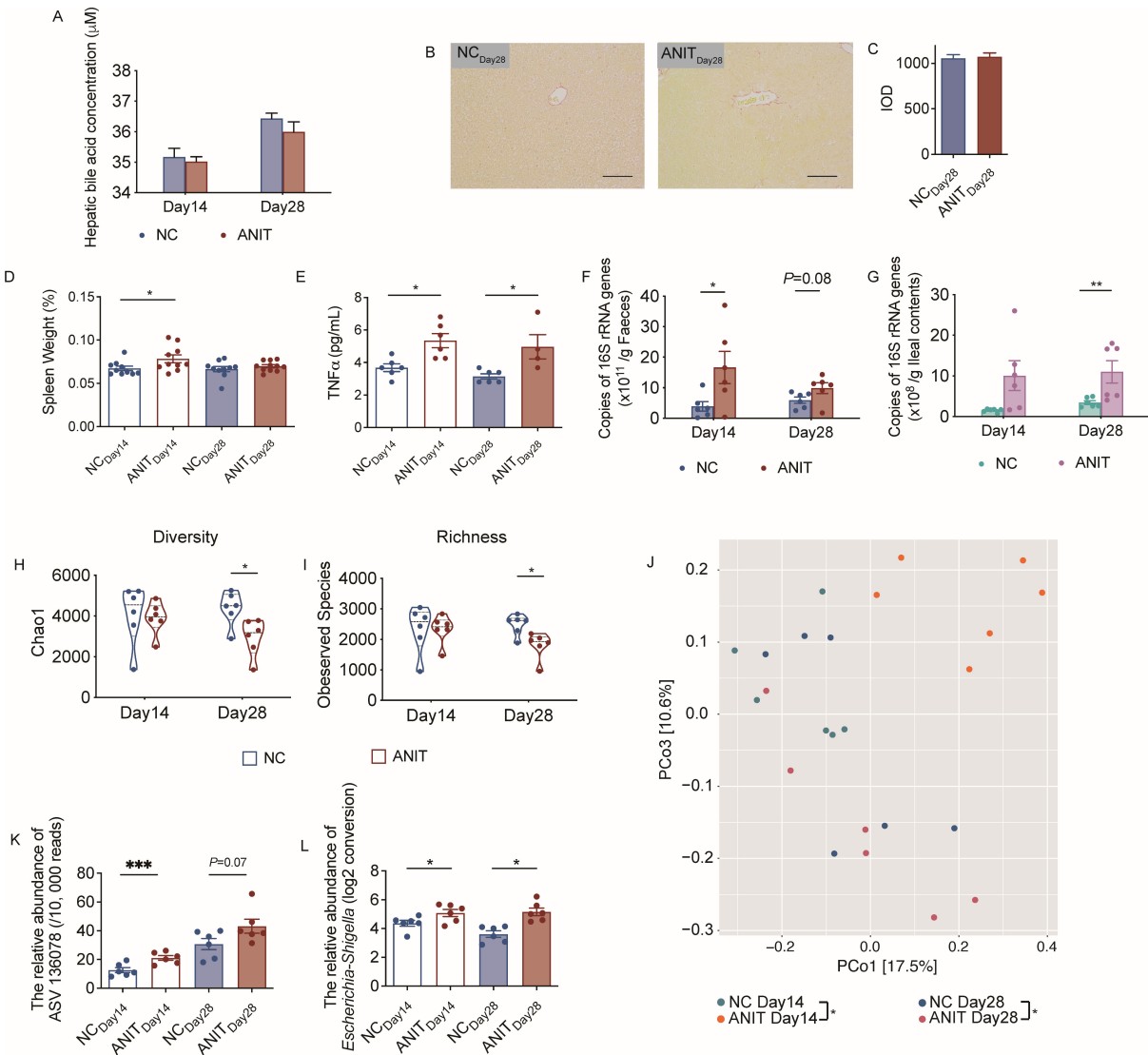

**FIG 3** The legacy effects of acute cholestasis-induced dysbiosis of gut microbiota. The experimental design is similar to Fig. 1A, but the mice were kept up to 28 days. (A) Hepatic bile acid concentration in the NC and ANIT groups on the day 14 and day 28, $n$ = 5–6/group. (B) Sirius Crimson-stained histological sections of hepatic tissue on the day 28 (400×, scale bar = 0.05 mm) and (C) IOD analysis of the liver fibrosis area from NC$_{Day28}$ and ANIT$_{Day28}$ mice, $n$ = 6/group. (D) Spleen weight and (E) serum TNFα concentration in the NC and ANIT groups on day 14 and 28, respectively; $n$ = 6/group. The bacterial load in the fecal sample (F) and the ileal content (G) of NC and ANIT mice on day 14 and day 28, respectively, as measured by real-time qPCR, $n$ = 6/group. (H) The alpha diversity and (I) richness (observed species) of gut microbiota in fecal samples on day 14 and day 28 from NC and ANIT groups; $n$ = 6/group. (J) Overall gut microbial structure in NC and ANIT mice on day 14 and day 28. PCoA was performed based on Bray-Curtis distance at the ASV level on day 14 and day 28; $n$ = 6/group. The relative abundance of ASV136078 (K) and *Escherichia-Shigella* (L) in the fecal samples of each group, $n$ = 6/group. NC$_{Day14}$, day 14 of the control mice; NC$_{Day28}$, day 28 of the control mice; ANIT$_{Day14}$, day 14 of the mice after 75 mg/kg ANIT gavage; ANIT$_{Day28}$, day 28 of the mice after 75 mg/kg ANIT gavage. The data in (A), (C), (D–I), (K), and (L) are expressed as mean ± SEM, and Student's $t$-test (two-tailed) was used to analyze differences between the following group pairs: NC$_{Day14}$ vs. ANIT$_{Day14}$, or NC$_{Day28}$ vs. ANIT$_{Day28}$. *$P$ < 0.05, **$P$ < 0.01 and ***$P$ < 0.001.

584 genes were potentially related to human diseases from the draft genome (Table S3), and 369 genes were potential virulence factor-coding genes (Table S4). Thus, the abundance of genes encoding virulence factors in the gut microbiota was significantly increased after cholestasis. The representative strain ASV136078 in the genus *Escherichia-Shigella*, which was increased in abundance after cholestasis, may carry many disease-related and virulence factor-coding genes. The CheckM calculation results for all bins can

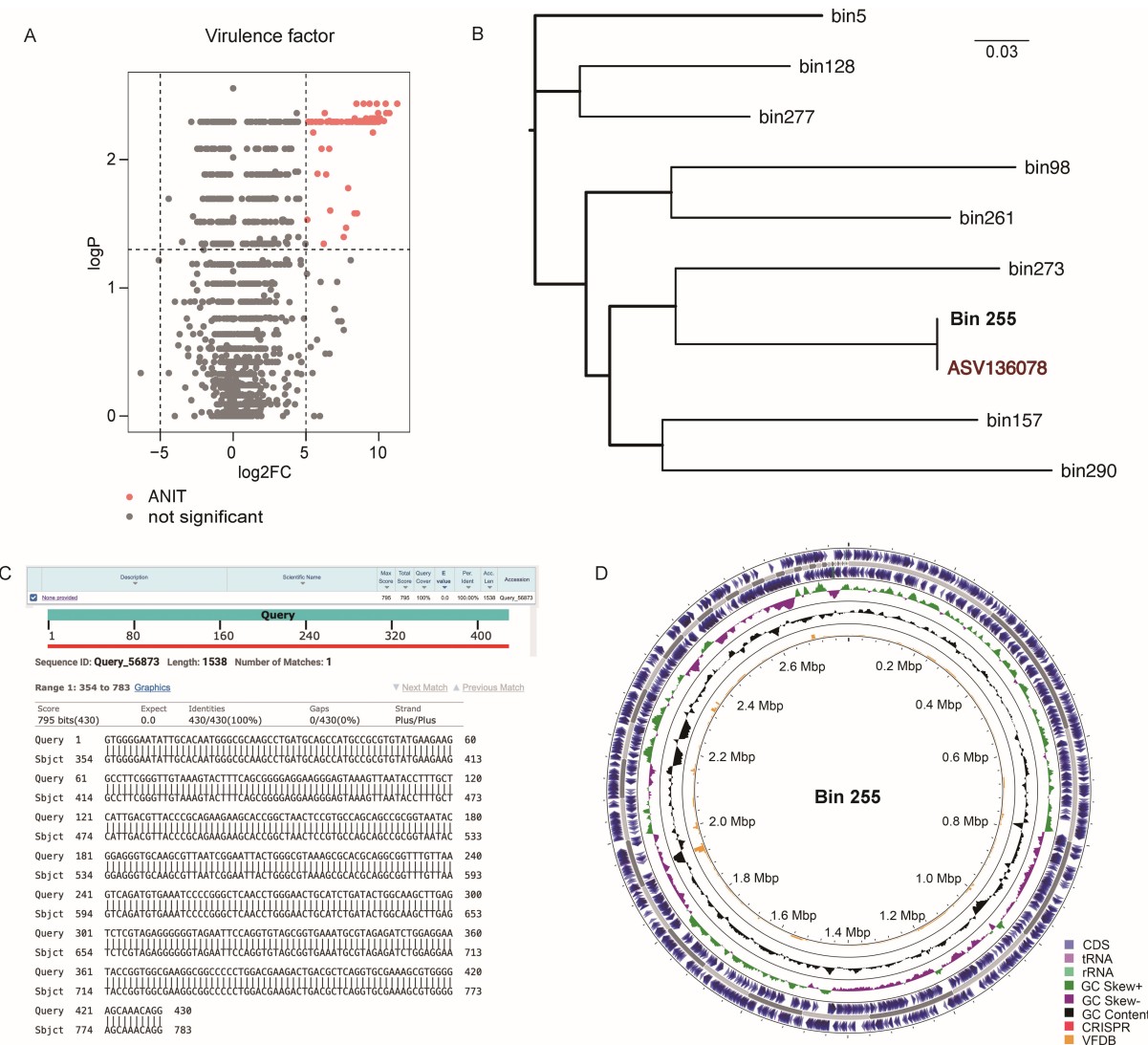

**FIG 4** ANIT-induced cholestasis increases gut microbial virulence factors encoding genes. (A) Virulence factors that were significantly enriched in ANIT-treated and control groups. The pink points in the volcano map represents the virulence factors that were significantly enriched in the ANIT group, and the specific names of virulence factors were shown in Table S2. (B) Phylogenetic trees constructed from ASV136078 and bins capable of extracting 16S rRNA genes. (C) Alignment results of 16S rRNA sequence of the draft genome of the strain (genus *Escherichia-Shigella*) with ASV136078. (D) Draft genome atlas of this strain of genus *Escherichia-Shigella*.

be found in Table S5. And the results of the species-level annotation and classification of the bins can be found in Table S6.

## Altered gut microbiota by cholestasis triggers hepatic inflammation in germ-free mice

To explore the *in vivo* function of gut microbiota altered by cholestasis, we transplanted fecal samples from ANIT-treated (day 2 after treatment) and control mice into germ-free C57BL/6J mice; the recipient mice were denoted as GF$_{NC}$ and GF$_{ANIT}$ mice, respectively. The fecal samples from the recipient mice were collected on day 8 after transplantation and sequenced. The gut microbiota structure in the GF$_{ANIT}$ group was significantly shifted away from the GF$_{NC}$ group, as shown by its reduced alpha diversity (Fig. 5 through C). We compared the PCoA score plots of Bray-Curtis distance based on the ASVs of the 16S rRNA gene V3-V4 region between donor and recipient samples. We found that the gut microbiota structure of the NC group mice was very similar to that

of the GF$_{NC}$ group, and the gut microbiota structure of the ANIT group mice was very similar to that of the GF$_{ANIT}$ group. This indicates that the gut microbiota structure of the donors was satisfactorily replicated in the recipients' intestines, demonstrating the success of the fecal microbiota transplantation experiment (Fig. 5D). There were also significant differences in the PCoA score plot of Bray-Curtis distance based on the ASVs of the 16S rRNA gene V3-V4 region ($P = 0.001$ by PERMANOVA test with 999 permutations, Fig. 5E). Quantification of the distances between PCoA score plots further demonstrated that the gut microbiota structures of the two donor groups were respectively replicated in the two recipient groups' intestines (Fig. 5F and G). In line with the findings from the ANIT-treated hosts, the genus *Escherichia-Shigella* and two ASVs (ASV136078 and ASV11534) were enriched in recipient GF$_{ANIT}$ mice (Fig. 5H and I). Next, we assessed the inflammation in the liver and found that the gene expression of *IL 1β*, *MCP-1*, *Timp-1*, and *aSma*, related to inflammation and injury, was significantly upregulated in GF$_{ANIT}$ mice (Fig. 5J). The expression of *IL 1β* in the colon of GF$_{ANIT}$ mice was also higher than that in the colon of GF$_{NC}$ mice (Fig. 5K). Concomitantly, gut luminal mucin genes (*muc1*, *muc2* and *muc3*), associated with intestinal permeability, were significantly upregulated in the colon of GF$_{ANIT}$ mice compared to GF$_{NC}$ mice (Fig. 5K), which may be related to the concentration of bile acids in the intestinal lumen. Moreover, the GF$_{ANIT}$ mice showed elevated hepatic bile acids (Fig. 5L), ALT (Fig. 5M), AST (Fig. 5N), and concentrations of inflammatory cytokines (Fig. 5O through Q), and all of which indicated the inflamed and damaged liver as the results of the transplantation of gut microbiota from the donor mice with cholestasis. Thus, altered gut microbiota mediated liver injury and liver inflammation in the germ-free hosts without administration of ANIT.

## DISCUSSION

Our study revealed that short-term hepatic cholestasis induces alterations of the small intestinal and colonic microbiota in both bacterial load and structural composition in mice. Furthermore, gut microbiota dysbiosis persists in a prolonged period following recovery from cholestasis, characterized by SIBO, reduced microbiota diversity, and an overall imbalance in composition, and all of which were accompanied by heightened hepatic inflammation. We found the genus *Escherichia-Shigella* in our cholestasis model, which is characterized by enrichment of genes encoding virulence factors and showed significant correlation with serum bile acids, exhibited a substantial increase in both the small intestine and colon. We found that many virulence factor-encoding genes in the spliced genome drafts of representative strains belong to the genus *Escherichia-Shigella*, suggesting that the high richness of virulence factor-encoding genes in fecal DNA might be the result of high abundance of *Escherichia-Shigella*. Moreover, fecal microbiota from the cholestatic mice-induced hepatic inflammation and disturbed bile acid metabolism in germ-free recipient mice indicating that cholestatically disturbed gut microbiota disrupts enterohepatic loop.

In our study, a decrease in secondary bile acids in the circulation indicated a fundamental alteration in the bile acid-degrading function of the gut microbiota in cholestatic states (37, 39). The inhibitory effect of bile acids on opportunistic pathogen growth was observed in different bacterial species including *Escherichia coli* (44), *Enterococcus fecalis* (44), *Clostridium scindens* (45, 46), *Clostridium hylemonae* (47), and *Clostridium hiranonis* (47). Some probiotic bacteria, such as *Lactobacillus*, *Bifidobacterium*, and *Bacillus*, resist the toxicity of bile acids by overexpressing a range of proteins devoted to the efflux of bile salts or protons (48, 49). We hypothesized that opportunistic pathogens sensitive to bile acids would increase in abundance in the absence of gut luminal bile acids. We performed high-throughput sequencing based on the V3-V4 region of the 16S rRNA gene of DNA from ileal contents and stool samples from cholestatic mice and assessed bacterial load by qPCR from DNA samples from cholestatic mice. We found an increased bacterial load in the intestinal lumen and an increased abundance of the genus *Escherichia-Shigella* represented by ASV136078, in ANIT-induced cholestatic mice. This indicates that cholestasis leads to SIBO, which is significantly

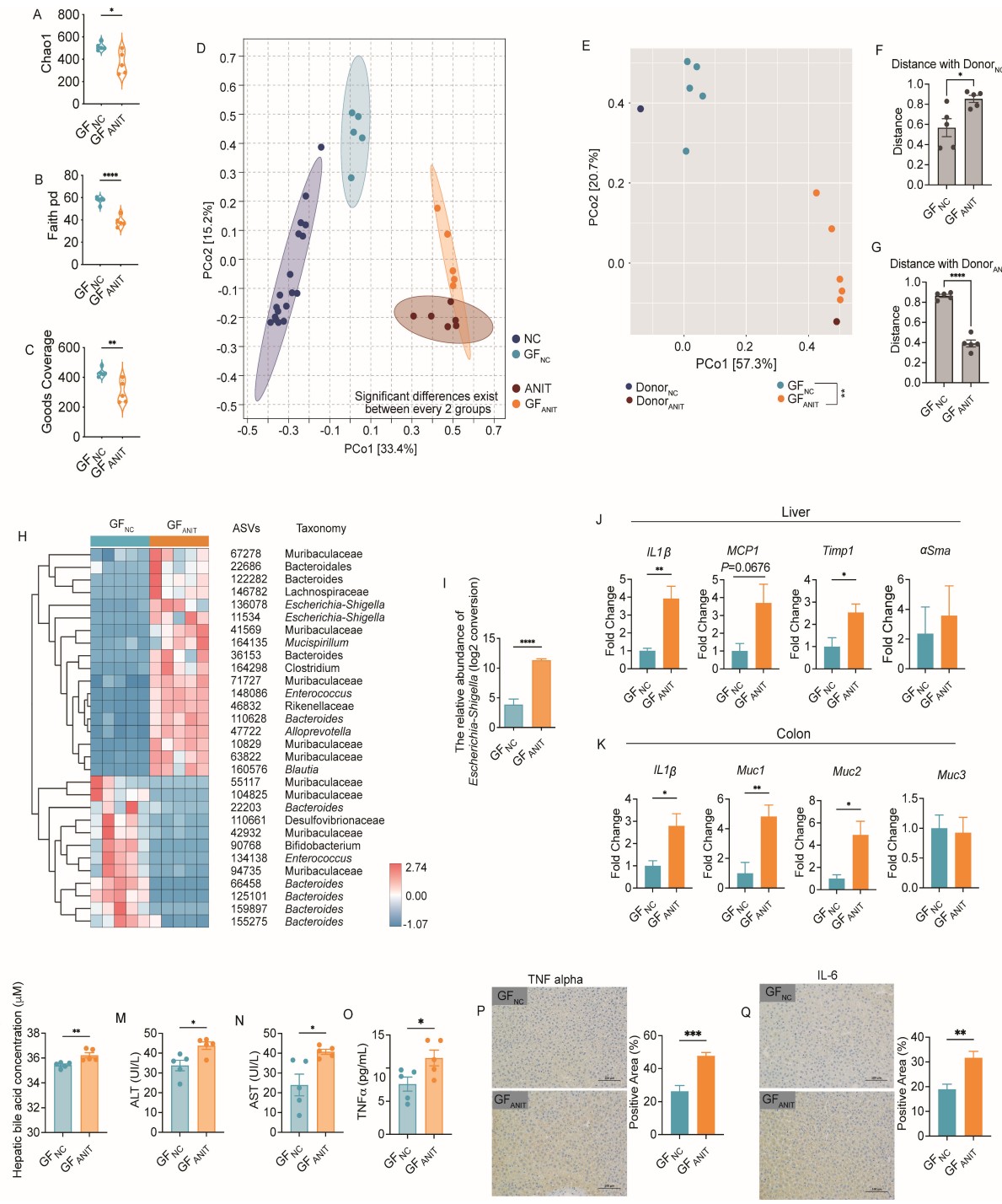

**FIG 5** Gut microbiota from cholestatic donors triggers hepatic inflammation in germ-free mice. Germ free C57BL/6 mice were transplanted with gut microbiota from cholestatic donors (day 2 after ANIT treatment) or the control donors (day 2 with water). GF_NC, the germ-free mice inoculated with the fecal microbiota from the mice of NC_Day2 group; GF_ANIT, the germ-free mice inoculated with the fecal microbiota from the mice of ANIT_Day2 group. The fecal samples (on the day 8) from the GF_NC and GF_ANIT mice were sequenced (A–C) The alpha diversity of gut microbiota of GF_NC and GF_ANIT group, $n = 5$/group. (D) Overall gut microbial structure in NC, ANIT, GF_NC, and GF_ANIT mice. PCoA was performed based on Bray-Curtis distance at the ASV level. (E) Overall gut microbial structure in Donor_NC, Donor_ANIT, GF_NC and GF_ANIT mice. PCoA was performed based on Bray-Curtis distance at the ASV level; $n = 5$/group. (F) The distances from Donor_NC to GF_NC and GF_ANIT, respectively. (G) The distances from Donor_ANIT to GF_NC and GF_ANIT, respectively. (H) Thirty ASVs that were significantly altered after transplanted the fecal microbiota of ANIT treated mice, as identified using random forest models. The cluster tree on the left shows associations between these ASVs, as determined by the Spearman correlation coefficient based on their relative abundances among all the samples. The heat map shows the relative abundance

**FIG 5** (Continued)

(log$_{10}$ transformed) of each ASV in a sample of each individual mouse; $n$ = 5/group. (I) The relative abundance of *Escherichia-Shigella* in the fecal samples of each group, $n$ = 5/group. (J) Levels of mRNA expression of *IL1β*, *MCP1*, *Timp1*, and *αSma* related with inflammation and hepatic injury in liver tissue on the day 8, $n$ = 5/group. (K) Levels of mRNA expression of *IL1β*, *Muc1*, *Muc2*, and *Muc3* related with inflammation and gut permeability in the colon tissue, $n$ = 5/group. (L) Total bile acids in the hepatic tissue, $n$ = 5/group. (M) Serum ALT and (N) AST and (O) TNFα concentrations; $n$ = 5/group. Immunohistochemistry staining of TNFα (P) and IL-6 (Q) of hepatic tissue sections (400×, scale bar = 0.05 mm) from GF$_{NC}$ and GF$_{ANIT}$ mice; $n$ = 5/group. Positive staining areas were calculated using Image Pro Plus software. The data are shown as mean ± SEM, and Student's *t*-test (two-tailed) was used to analyze differences GF$_{NC}$ vs. GF$_{ANIT}$. *$P$ < 0.05, **$P$ < 0.01 and ***$P$ < 0.001.

associated with coagulopathy (50), venous thromboembolism (51), hyperammonemic encephalopathy (52), and pancreaticobiliary disorders (53).

Most patients with non-hereditary cholestasis recover with timely intervention and advanced therapies (6). However, temporal changes in intestinal microbiota in the intestinal lumen, including the ileum and colon, have not been studied and evaluated after cholestasis has healed. We found that during ANIT treatment, dysbacteriosis did not recover and the relative abundance of *Escherichia-Shigella* remained high after the liver indicators returned to normal. The results of shotgun metagenomic sequencing showed that the cholestatic intestinal microbiota contained a higher abundance of virulence factor-encoding genes, and our draft genome of ASV136078 indicated that this strain of *Escherichia-Shigella* bacteria also contained many virulence factor-encoding genes. Prior research has suggested that certain opportunistic gut pathogens possessing higher virulence factors are typically more sensitive to the presence of bile acids, whereas some probiotic strains exhibit greater resistance to bile acids (44, 48, 49). Our findings indicate that bile acids and cholestasis influence the composition of the gut microbiome, potentially affecting the abundance of intestinal pathogens. This suggests a possible role for cholestasis in altering microbial dynamics, although direct evidence linking bile acid levels to specific changes in the microbiome is yet to be established. Further research is needed to elucidate the mechanisms involved.

We substantiated the causal role of the cholestatic-disturbed gut microbiota in provoking liver inflammation and disrupting bile acid metabolism through gut microbiota transplantation experiments in germ-free mice. Previous studies demonstrated the presence of intestinal microbiota-derived enzymes responsible for the degradation and modification of PBAs (37). Consequently, transplanting intestinal microbiota into the second hosts can, to some extent, replicate the donor's bile acid metabolism profile, subsequently influencing overall organismal health (54–57). The observed biological effects on the intestinal microbiota during cholestasis suggest that restoration of the microbiota may be an important consideration in the management of the disease. Therefore, adjuvant therapies aimed at targeting the gut microbiota may be necessary to promote full recovery in patients with cholestasis.

Our study provides new insights into the impact of acute hepatic cholestasis on the structural composition of the small intestinal and colonic microbiota and their subsequent effects on hepatic inflammation. We acknowledge several limitations of our research. Firstly, despite support from the published study (58) and our current data in studying the role of gut microbiota in liver injury using ANIT-induced acute cholestasis and liver injury model, we cannot completely exclude the direct impact of ANIT, with oral administration, on the gut microbiota. This might limit our understanding of the precise role of gut microbiota in liver inflammation and bile acid metabolism. Secondly, our model is limited to mice and may not fully reflect the human condition. Therefore, future research needs to validate our findings in different biological model systems and explore the altered microbiota and their role in human cholestatic settings. Lastly, while we validated the impact of gut microbiota on hepatic inflammation and bile acid metabolism through gut microbiota transplantation experiments in germ-free mice, this approach may not fully mimic the complex human gut environment. Thus, further research is needed to explore the specific mechanisms, by which gut microbiota regulates bile acid metabolism and liver function in humans. By recognizing the

limitations, we hope to promote future research in our future studies and the studies by the scientists in the field.

Overall, our findings have important implications for understanding the gut microbiome in the context of cholestasis and may provide a rationale for promoting continued monitoring gut microbiota and microbiota-targeted interventions in cholestatic patients.

## ACKNOWLEDGMENTS

This work was supported by The International Postdoctoral Exchange Fellowship Program of The Office of China Postdoctoral Council (No. 2021044), National Natural Science Foundation of China (No. 82004149) and China Postdoctoral Science Foundation No. 2018M630465 to and the Shanghai Collaborative Innovation Center for Chronic Disease Prevention and Health Services (2021 Science and Technology 02-37) to X.Y, R.R.W and L.Z. The National Institutes of Health HD 097808, DK 126809, DK 130318, Diabetes Action Research and Education Foundation to L.W.

X.Y. and Y.S.X. analyzed the data and conducted the experiments. X.Y., R.R.W. and L.W. prepared and edited the manuscript. X.Y., R. R. W., G. J. and L.W. designed the experiments. J.L., X.M.R, Z.H.G, L.F.S, and L.Z. conducted some experiments. X.Y. performed the bioinformatics analysis on microbiology. R.R.W. and G.J. supervised this study. R.R.W. and G.J. conceived of the study. R.R.W., G.J. and L.W. are the guarantors of this work and, as such, had full access to all the data in the study and take responsibility for the integrity of the data and the accuracy of the data analysis.

## AUTHOR AFFILIATIONS

[1]Shanghai Innovation Center of TCM Health Service, Shanghai University of Traditional Chinese Medicine, Shanghai, China
[2]Department of Food Science and Technology, School of Agriculture and Biology, Shanghai Jiao Tong University, Shanghai, China
[3]Section of Endocrinology, Internal Medicine, School of Medicine, Yale University, New Haven, Connecticut, USA
[4]Department of Biostatistics and Bioinformatics, Department of Biostatistics and Bioinformatics, Rollins School of Public Health, Emory University, Atlanta, Georgia, USA
[5]School of Materials Science and Engineering, East China University of Science and Technology, Shanghai, China
[6]General Medicine, Medical school, Kunming University of Science and Technology, Kunming, China
[7]Institute of Digestive Diseases, Longhua Hospital, Shanghai University of Traditional Chinese Medicine, Shanghai, China

## AUTHOR ORCIDs

Xin Yang ⓘ http://orcid.org/0000-0001-7799-7425
Li Wen ⓘ http://orcid.org/0000-0001-7799-7425
Guang Ji ⓘ http://orcid.org/0000-0003-0842-3676
Ruirui Wang ⓘ http://orcid.org/0000-0002-2766-9420

## FUNDING

| Funder | Grant(s) | Author(s) |
| --- | --- | --- |
| The International postdoctoral Exchange Fellowship | 2021044 | Xin Yang |
| HHS \| National Institutes of Health (NIH) | HD 097808, DK 126809, DK 130318 | Li Wen |
| MOST \| National Natural Science Foundation of China (NSFC) | No. 82004149 | Ruirui Wang |

## AUTHOR CONTRIBUTIONS

Ximing Ran, Investigation | Li Wen, Conceptualization, Data curation, Formal analysis, Funding acquisition, Investigation, Validation, Visualization, Writing – original draft, Writing – review and editing.

## DATA AVAILABILITY

The sequenced datasets generated during the current study are available in the NCBI repository (PRJNA975699 and PRJNA974219).

## ADDITIONAL FILES

The following material is available online.

### Supplemental Material

**File S1 (mSystems00127-24-s0001.fasta).** The 16S rRNA gene of ASV136078.
**File S2 (mSystems00127-24-s0002.fasta).** The draft genome of ASV136078.
**Supplemental Figures (mSystems00127-24-s0003.pdf).** Figure S1-S6.
**Supplemental Legends (mSystems00127-24-s0004.docx).** Supplemental figure, table, and file legends.
**Table S1 (mSystems00127-24-s0005.xlsx).** Primer list for qPCR.
**Table S2 (mSystems00127-24-s0006.xls).** Abundance matrix of genes encoding virulence factors in fecal samples.
**Table S3 (mSystems00127-24-s0007.xls).** Disease-related coding gene information annotated in the draft genome.
**Table S4 (mSystems00127-24-s0008.xls).** Information on encoding virulence factors genes annotated in the draft genome.
**Table S5 (mSystems00127-24-s0009.xls).** The CheckM calculation results for all bins.
**Table S6 (mSystems00127-24-s0010.xls).** The results of the species-level annotation and classification of the bins.

### Open Peer Review

**PEER REVIEW HISTORY (review-history.pdf).** An accounting of the reviewer comments and feedback.

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
