## [Reviewer comments · mSystems]

Bile Acid-Gut Microbiota Imbalance in Cholestasis and Its Long-term Effect in Mice

Xin Yang, Yuesong Xu, Jie Li, Zhihao Gu, Linfeng Song, Lei Zhang, Li Wen, Ruirui Wang, and Guang Ji

Corresponding Author(s): Li Wen, Yale University School of Medicine

Review Timeline:

Submission Date:	January 26, 2024
Editorial Decision:	March 14, 2024
Revision Received:	May 9, 2024
Accepted:	May 30, 2024

Editor: Daniel Garrido

Reviewer(s): Disclosure of reviewer identity is with reference to reviewer comments included in decision letter(s). The following individuals involved in review of your submission have agreed to reveal their identity: Anand Kumar (Reviewer #1); Sudip Das (Reviewer #2); Kazuyuki Kasahara (Reviewer #4)

Transaction Report:

DOI: <https://doi.org/10.1128/msystems.00127-24>

Re: mSystems00127-24 (Bile Acid-Gut Microbiota Imbalance in Cholestasis and Its Long-term Effect in Mice)

Dear Dr. Li Wen:

While reviewers found your manuscript of great interest and potential, there are still some significant issues to be addressed.

Revision Guidelines

Sincerely,
Daniel Garrido
Editor
mSystems

Reviewer #1 (Comments for the Author):

The present study aims to explore the involvement of gut microbiota in cholestasis and its short- and long-term effects. The authors observed disrupted microbiota in an animal model of cholestasis, alongside typical symptoms such as liver inflammation, damage, and altered bile acid production. Interestingly, they found that mice receiving fecal transplants from cholestatic mice developed similar symptoms, emphasizing the role of gut microbiota in cholestasis. This study underscores the potential for targeting gut microbiota to mitigate the associated negative consequences.

Major comments:

1. The reviewer is interested in comparing the phenotypic data acquired from ANIT versus GF-ANIT, as well as NC versus GF-NC. Since the authors have already gathered and analyzed this data, they can conduct such a comparison and present the results. This will bolster the evidence supporting the role of gut microbiota in cholestasis.
2. Additionally, outlining a comprehensive hypothetical model elucidating how gut microbiota may contribute to cholestasis and its detrimental effects would be an effective way to captivate the readers' attention.

Minor comments:

- Line 76-78: Sentence is not clear, re-word it.
- Line 105: Expand IVC system.
- Line 111: SLAC stands for what?
- Line 113 to 115: Can be deleted as described previously.
- Line 116: by the-repeated twice, delete one.
- Line 119: Indicate number of animals in short term experiment too.
- Line 135 to 136: Why second day fecal ample was collected just describe briefly here.
- Line 143, 146 and 149: While kits were utilized to ascertain the concentration in question, the authors aim to provide a concise description of the process involved and any deviations encountered during its execution.
- Line 299: fully recovered on what day?
- Figure are duplicated at the end.

Reviewer #2 (Comments for the Author):

Please find the review in the attached word file.

Reviewer #3 (Comments for the Author):

Title: Bile Acid-Gut Microbiota Imbalance in Cholestasis and Its Long-term Effect in Mice

In this study, the authors characterized host phenotypes and the microbiome in an induced short-term hepatic cholestasis mouse model induced by α -Naphthyl-isothiocyanate (ANIT). First, the authors did a thoughtful confirmation of acute liver injury in mice by ANIT. They also showed a different microbial composition between mice treated with ANIT and control mice by characterizing the small intestine content and fecal samples using 16S rRNA amplicon sequencing. Specifically, the authors identified a correlation between *Escherichia-Shigella* and certain bile acids, including secondary bile acids. Then, the authors showed the persistence of various pathologies in non-hepatic organs and tissues over a more extended period in their model, including systemic inflammation by higher levels of circulating TNF. Regarding the microbiome, the authors showed a persistent bacterial load in fecal and ileal content. Interestingly, germ-free mice colonized with fecal samples for mice treated with ANIT trigger damage and hepatic inflammation. This host response is not observed in germ-free mice colonized with fecal samples for control mice.

This is a relevant study that established a simple mice model system to determine interactions between cholestasis and the gut microbiome compared to other systems, such as bile duct ligation. Using this model, the authors showed an unexpected persistence of pathologies on non-hepatic organs and tissues and dysbiosis over a long period after the self-limiting periods, underlining the relevance of recovery periods to manage the disease. However, I have some concerns about the microbiome data analysis.

Comment 1.

The authors argue that there is no change in the control group's microbiome structure over time and decided to merge all the control samples for statistical analysis (Lines 340-342). However, looking at the PCo plot in Fig 2C, I disagree with the authors. There is a variability of the control samples over the second coordinate in the PCo. By analyzing the PCo plot, I believe that the ANIT treatment induces the development of a different microbiome state. Still, I do not consider the best strategy to merge all the control data for statistical analyses. Please provide a strong argument for this decision or consider doing all the statistical analyses without merging the control samples.

In the mice experiments, the authors randomize the selection of mice to create two groups: (i) normal control (NC) group and (ii) ANIT group. This strategy should avoid bias in the starting mice microbiome, such as maintaining mice in different cages before the experiment. According to Fig S3, the authors collected fecal samples at time 0 and processed those samples. It is important to show the microbiome data of time 0, in for example Fig 2A and Fig S3A-D, to confirm that mice for both groups started with a similar microbiome and that the observed microbial dysbiosis in the ANIT group corresponds to the treatment.

Comment 2.

There is an unusual pattern on day 4 in the short-term experiments. The alpha diversity values of the control sample (Fig 2A and 2B) on day 4 seem lower than those on days 2 and 8. In addition, the food intake on day 4 for the control sample is lower than the other days. As a control, I would not expect variability in the samples. Are Chao1 and Observed species on day 4 statistically different from days 2 and 8? If so, do you have any idea of this variability? In the material and method section, it says that this experiment was repeated one time. Is this data a mix of both experiments?

Comment 3.

The current Fig S4 is of low quality. Please upload a figure with a better resolution.

Comments 4.

In Fig 2A and 2B, the authors showed a recovery of the alpha diversity metric at day 8 in the ANIT group. However, the alpha diversity metric decreased again on Day 14 and Day 28. Is this correct? If so, do you have any hypothesis of this dynamics (recover and then collapse of the richness)?

Comments 5.

For the germ-free experiments, the authors showed a dissimilarity of microbiomes between the one recovered from germ-free mice inoculated with feces from control mice and the one recovered from germ-free mice inoculated with feces of ANIT mice (Fig 5A-D). However, the authors also need to show that the microbiome engrafted in the corresponding germ-free mice is similar to the original inoculum to confirm whether the engraftment was successful or not.

Comment 6.

In the discussion section (Line 518-521), the authors argue, "Our observation suggests a critical role for bile acids in reducing the abundance of intestinal pathogens and raise the possibility that cholestasis may be a significant contributor to the increase in opportunistic pathogen abundance." This study highlights a broad physiological effect in the ANIT mice, including non-hepatic diseases and not only changes in bile acids. The authors indeed showed the impact of cholestasis in gut microbiome composition, but there is no evidence of direct role of bile acid in changing the microbiome. I would consider tone-down this sentence.

Comment 7.

The important section is missing.

Reviewer #4 (Comments for the Author):

The authors assessed gut microbial dysbiosis and bile acid profiles in a drug-induced cholestasis mouse model. The paper not only analyzed the phenotype and microbiome of ANIT-treated mice but also conducted the analysis of genes encoding microbial virulence factors and tried to establish causality using germ-free mice. These aspects merit commendation. However, a significant concern in this paper is the adoption of an oral administration model for the drug (ANIT) despite focusing on gut microbial dysbiosis in cholestasis. This is problematic as it cannot rule out the direct impact of the drug on the microbiome (indeed, the possibility is quite high) and is not an ideal model for studying the relationship between the disease and dysbiosis. Moreover, they have not clearly stated the hypothesis and/or objectives of this study, making it unclear what the specific research aims are. Below is a list of specific comments from the reviewer.

1. As mentioned earlier, the oral administration model of the drug (ANIT) used in this manuscript is not suitable for studying the relationship between the disease and dysbiosis because the direct impact of ANIT on the microbiome is considered substantial.
2. The relative abundance of *Escherichia-Shigella* is presented in Fig 2F and Fig 3L. However, the values for each control group differ significantly. Please provide an explanation for this discrepancy.
3. Fig 2L indicates an increase in *Escherichia-Shigella* in the small intestine, but how did other taxa change?
4. It is surprising that in the germ-free model experiment, administering faeces without the drug had an impact on liver function and inflammatory cytokines. What is the reason for intentionally not administering ANIT? Additionally, details of this experiment are not outlined in the methods; for example, when was the analysis conducted after the faecal transplantation?
5. In the same experiment (Fig 5), they examined the gene expression of *muc1*, *muc2*, and *muc3* as mucin-related genes. While the importance of *muc2* in mucin layer formation in mice is well-known, explain the rationale behind investigating *muc1* and *muc3*.

6. In the same experiment (Fig 5), was the bacterial community in the small intestine affected? For example, was there any evidence of small intestine overgrowth of bacteria?

Msystems00127-24 Review:

In this study Yang et al. show the host and gut microbiota axis in case of Cholestasis and how altered commensal dynamics and increased abundance pathogenic bacteria induces altered hepatic function and inflammatory responses. This study is well done with attention to details and is methodologically strong especially the usage of fecal microbiome transplantation and germ-free animals makes a strong case for causal effects. The study is already strong and with modifications and additional data suggested, this will end up of remarkable quality.

Please find my comments below:

Minor comments:

1. Please check for typos in the abstract.
2. I suggested a change in vocabulary in the methods part in:

“**Long-term experiment:** After one-week of acclimation, the mice were randomly assigned to two groups: (i) NC group (n = 12), control mice had drinking water for 28 days; (ii) ANIT group (n = 13), mice were administered with 75 mg/kg ANIT (Sigma-Aldrich, St. Louis, MO, USA) through oral gavage for 28 days. “

3. Please make the titles more descriptive of the results that methodological statements.
4. Please do not introduce an axis breaks in gene expression instead choose to use log2-fold which will produce much better representation.
5. Please introduce to the abbreviations like PBA and SBA in the text and figure legends.

Major comments:

1. Data availability and reproducibility: Please write the full github address as hyperlink to be transferred to the website when needed. Please provide all processed data (except for raw reads and codes) as a zenodo repository <https://zenodo.org> and link Github to it.
2. The authors didn't I use CheckM during assembly. Considering the importance of CheckM in assessing the quality of assembled genomes and identifying potential contamination, the authros are encouraged to use it. Additionally, how do the authors ensure the reliability of the assembled genomes and the accuracy of downstream analyses?
3. What is the justification for using male animals and not a mixed population? Does Cholestasis primarily occur in male population?
4. Please elaborate on the histopathological methods as it is not clear in the methods part. How was the scoring done and was it a single / double not blinded scoring or was it performed by a trained mouse pathologist. Also please explain the scoring system to assess tissue damage.

5. The authors have an excellent set of data from different related microbial niches, and it would be a missed opportunity not describing each microbiota profile properly. Although, the authors have made commendable efforts on presenting their data but for exploratory purpose of the microbiota, it would be informative to construct prevalence and abundance plots of each microbial niche at genera / species in Figure 2. As a reader, I would appreciate such descriptive plots and will provide me with the idea on which species are the primary members in gut.
6. In Result / Figure 2, the authors didn't choose to use the conventional Shannon and Simpson index in the main figure. It is important to show at least the Simpson diversity index should be used. This is a mathematical measure used in ecology to quantify the diversity or richness of species in each community or ecosystem. It provides insights into how evenly individuals are distributed among different species in a community. A higher Simpson diversity index indicates lower diversity because it means that there is a higher probability that two randomly selected individuals will belong to the same species. In this study the Simpson index is higher in NC, which is counter intuitive. In general, there are two ways one can calculate this:

i. Simpson's Index (D) = $1 - \sum(p_i^2)$

where:

- **D** is the Simpson diversity index.
- **p_i** is the proportion of individuals belonging to the i-th species.

This index quantifies the dominance of a few dominant species in a community.

ii. Hill Numbers (Diversity Order q):

Hill numbers provide a family of diversity indices that can be tailored to different research questions by varying the diversity order "q."

If the authors are primarily interested in understanding dominance and the influence of a few dominant species in your community, Simpson's index (D) might be more appropriate. If you want to consider both the richness and evenness of species in your community and have the flexibility to emphasize one aspect over the other, Hill numbers with different "q" values can be a powerful choice.

Please indicate which method was used due to contradictory results.

7. Regarding Result / Figure 2, the authors use Chao1 index. It is arguably a flawed method of indicating species richness. It is sensitive to number of taxa appearing in each sample and sequencing depth achieved. This means that two samples when subjected to the same sequencing effort may produce different richness values. Are the alpha diversity hill indices? Shannon and Simpson indices are more robust and should be used here. Please use the Renyi-Hill number calculation in vegan package for this (Hill numbers 1 to 4). Please note that Shannon diversity in other packages such as *phyloseq* and QIIME is not the same as Hill indices, which are more comparable between each other. For help follow these articles:

<https://esajournals.onlinelibrary.wiley.com/doi/10.1890/13-0133.1>
<https://www.nature.com/articles/srep38263>

8. The authors chose to use Bray-Curtis dissimilarity for beta diversity, which is a straightforward metric based on the relative abundances of taxa and does not consider phylogenetic relatedness. However, performing UniFrac analysis that considers species phylogeny will provide a much higher resolution data on the contribution to the disease/condition. This becomes more important since the authors observed changes in microbiota diversity with lung adenocarcinoma. UniFrac incorporates phylogenetic information, which means it considers the species and evolutionary relatedness. UniFrac distances are robust to differences in sequencing depth, making them suitable for comparing samples with varying levels of sequencing coverage. UniFrac distances are more biologically meaningful because they reflect the genetic divergence between microorganisms. I would suggest performing this alongside BC-measure and report pairwise PERMANOVA between conditions.
9. In result / figure 3, were these groups done in a separate experiment or together with the experiments performed for Figure 1 and 2. If not then did the authors look for similar effects on Day 2 and Day 8 using basic measurements? which would be consistent with the observations with Day 14 and Day 28.
10. In result / figure 3, why did the authors choose to separate Day 2 - Day 8 and Day 14- Day 28 and not put it in the same comprehensive microbiota analysis as Figure 2? Why wasn't TNFa measure in Figure 2?
11. In result / figure 3, the authors should elaborate in the methods more on the method used for constructing this genome as it is not clear. Were MAGs constructed for this? Instead of sequence matching ASV136078 the authors should use pipeline like MetaPhlan or Kaiju that will provide community data as well as metagenome data. Then these metagenomes can be used to reconstruct MAGs for E. coli / Shigella. Performing a more metagenome-based phylogeny will also remove the ambiguity of whether it is a Shigella or E. coli. The use of multiple tools is encouraged. Upon successful MAGs one can now produce phylogenetic trees to locate these bacteria from an evolutionary stand point.
12. The authors here indicate that it is the same ASV of E.coli-Shigella here. Were these sequencing analysis performed together with the ones from previous figure? The authors here choose to perform amplicon sequencing in this case as well. This is of course needed for a global picture. However, after performing metagenomic based identification of Shigella / E. coli and reconstruction of genomes. The authors should choose a different marker than 16S rRNA and a gene from E. coli Shigella genome that they have got to precisely identify if it is the same genotype that came up in the previous experiments. This should be used to perform unique-gene based amplicon sequencing.
13. Why faecal microbiome transfer in short term experiment not in long term? Did the authors perform long term experiments to investigate legacy effects. This effect must be shown to claim long-term effects in the article title.
14. It is not clear if pro-inflammatory cytokines were also high upon gut dysbiosis (colonic) measured from fecal samples except for Figure 5. Although the authors show ileal and hepatic expression of inflammatory signals, this lacks for Figure 2, which establishes the foundation of this article. In addition, the random forest models can used to indicate the bacterial species associated with both hepatic and colonic

gene expression. This becomes more important since one would expect a case of bacterial diarrhea. I would urge the authors to show this if possible to make a strong case of collateral effects of gut dysbiosis.

We would like to thank the reviewers for their time and effort on providing critical comments to our manuscript. Below is our point-by-point response to the comments.

Reviewer #1 (Comments for the Author):

The present study aims to explore the involvement of gut microbiota in cholestasis and its short- and long-term effects. The authors observed disrupted microbiota in an animal model of cholestasis, alongside typical symptoms such as liver inflammation, damage, and altered bile acid production. Interestingly, they found that mice receiving fecal transplants from cholestatic mice developed similar symptoms, emphasizing the role of gut microbiota in cholestasis. This study underscores the potential for targeting gut microbiota to mitigate the associated negative consequences.

We thank the reviewer for the positive feedback and encouragement.

Major comments:

1. The reviewer is interested in comparing the phenotypic data acquired from ANIT versus GF-ANIT, as well as NC versus GF-NC. Since the authors have already gathered and analyzed this data, they can conduct such a comparison and present the results. This will bolster the evidence supporting the role of gut microbiota in cholestasis.

We reanalyzed the data according to the suggestion and the results indeed further support the role of gut microbiota in cholestasis. We have added this set of comparison in the revision as **Fig.5D**. We used the pooled fecal samples from NC and ANIT donors, respectively, as the fecal microbiota transfer (FMT) source and colonized germ-free (GF) mice. The results of the Principal Coordinates Analysis (PCoA) scatter plot clearly showed that the microbial community structure of the GF_{NC} recipient group was similar to that of its donor, and similarly, the microbial community structure of the GF_{ANIT} recipient group was similar to that of its donor (**Fig. 5E**). By quantifying the distance among points on the PCoA scatter plot, we also

found that the gut microbiota structure of the recipient mice that received fecal transplants was significantly closer to their respective donor's microbiota structure (**Fig. 5F** and **Fig. 5G**). These data indicate that the fecal microbiota from ANIT donors transferred the structural characterization of the gut microbiota of donor mice. We also compared the gut microbiota structure between donors and their corresponding recipients. Although there are differences in the gut microbiota between donors and recipients, the gut microbiota structure of donor mice from the NC group is very similar to that of recipient mice transplanted with fecal samples from the NC group, and the gut microbiota structure of donor mice from the ANIT group is very similar to that of recipient mice transplanted with fecal samples from the ANIT group. We have added this set of comparison in the revision as new **Fig. 5D**. We have revised **Fig. 5** and the corresponding figure legends.

2. Additionally, outlining a comprehensive hypothetical model elucidating how gut microbiota may contribute to cholestasis and its detrimental effects would be an effective way to captivate the readers' attention.

We thank the reviewer's suggestion and create a graphical abstract (see below, also in the revised manuscript) to facilitate the understanding how gut microbiota may contribute to cholestasis and its detrimental effects. We hope this graphical abstract will be helpful for the readers.

Fig. R1 | Graphical Abstract.

Minor comments:

- **Line 76-78: Sentence is not clear, re-word it.**

We have revised this sentence to "An imbalance in the gut microbiota, partially driven by bile acid metabolism, may play a role in the development of various diseases in the host".

- **Line 105: Expand IVC system.**

We provided the full name of IVC in the revision. IVC stands for the Individually Ventilated Cage (IVC) system, a modern technology used in laboratory animal facilities, particularly for housing mice and other small rodents. The IVC system is designed to maintain a controlled environment for each cage, enhancing the welfare of the animals and reducing the risk of contamination between cages.

- **Line 111: SLAC stands for what?**

SLAC stands for Shanghai Laboratory Animal Center, where the germ-free mice were bred.

- **Line 113 to 115: Can be deleted as described previously.**

We have deleted this sentence and revised as "The mice were maintained under the same standard conditions".

- **Line 116: by the-repeated twice, delete one.**

We have deleted this redundant 'by the'.

- **Line 119: Indicate number of animals in short term experiment too.**

We have indicated the number of mice used in the short-term experiment.

- **Line 135 to 136: Why second day fecal sample was collected just describe briefly here.**

Day 2 was the time when all the hepatic phenotypes became evident as shown in **Figure 1B to L**. Gut microbiota were also significantly altered on day 2 shown in **Figure 2**. We, therefore, collected the fecal samples from this time point and used them as the source for the fecal microbiota transplantation (FMT) experiments.

- **Line 143, 146 and 149: While kits were utilized to ascertain the concentration in question, the authors aim to provide a concise description of the process involved and any deviations encountered during its execution.**

The tests were controlled by using the standard curve from recombinant ALT and AST. We have added this in the revision. Due to the word limit of the journal, we could not add more words and we hope that our revised brief description of quality controls will meet the reviewer's satisfaction.

- **Line 299: fully recovered on what day?**

By Day 4, the food and water consumption of ANIT treated mice had returned to normal, i.e., no significant difference from the control group, and the accumulation of bile acids in the liver had completely resolved by Day 8 (**Fig.1K**). Although the levels

of ALT and AST were significantly reduced on Day 8, they were still higher than the control mice (**Fig.1D & E**), so did serum inflammatory cytokine TNF α (**Fig.1F**). The concentration of bile acids in the fecal samples and serum of the ANIT treated mice also showed significant differences compared to the control group (**Fig.1L** and **Fig.1M**). If the level of hepatic bile acids was considered as a criterion for assessing recovery from cholestasis, thus, Day 8, would be the day when the mice had recovered.

• **Figure are duplicated at the end.**

To facilitate easy read, we had included figures in the manuscript. This allows each figure and its corresponding legend to be viewed together. We also uploaded each figure separately, which may result in duplicates. We have corrected this.

Reviewer #2 (Comments for the Author):

In this study Yang et al. show the host and gut microbiota axis in case of Cholestasis and how altered commensal dynamics and increased abundance pathogenic bacteria induces altered hepatic function and inflammatory responses. This study is well done with attention to details and is methodologically strong especially the usage of fecal microbiome transplantation and germ-free animals makes a strong case for causal effects. The study is already strong and with modifications and additional data suggested, this will end up of remarkable quality.

We are grateful for the reviewer's positive feedback and encouragement.

Please find my comments below:

Minor comments:

1. Please check for typos in the abstract.

We have added hyphen to "post-recovery" for grammatical correctness. We also changed "a cholestasis mice" to "cholestasis mice".

2. I suggested a change in vocabulary in the methods part in :“Long-term experiment: After one-week of acclimation, the mice were randomly assigned to two groups: (i) NC group (n = 12), control mice had drinking water for 28 days; (ii) ANIT group (n = 13), mice were administered with 75 mg/kg ANIT (Sigma-Aldrich, St. Louis, MO, USA) through oral gavage for 28 days.”

We have revised.

3. Please make the titles more descriptive of the results that methodological statements.

We have revised the title in the 1st Results section from "Acute liver injury induced by cholestasis in α -Naphthyl-isothiocyanate (ANIT) treated mouse model" to "Mouse Model of Cholestasis-Induced Acute Liver Injury and Bile Acid Redistribution by α -Naphthyl-isothiocyanate (ANIT)". We think that the other titles in the Results section adequately summarize the contents.

4. Please do not introduce an axis breaks in gene expression instead choose to use log₂-fold which will produce much better representation.

We revised **Fig. 1I** and **Fig. 1J**.

5. Please introduce to the abbreviations like PBA and SBA in the text and figure legends.

We have provided the full names in the revision.

Major comments:

1. Data availability and reproducibility: Please write the full github address as hyperlink to be transferred to the website when needed. Please provide all processed data (except for raw reads and codes) as a zenodo repository

<https://zenodo.org> and link Github to it.

According to the requirement by the journal on data availability (<https://journals.asm.org/availability-materials-and-data>), we have uploaded all the raw data to the NCBI database. We have set the database access link as a hyperlink in the revised manuscript. Additionally, we have uploaded the processed data to ZENODO as the reviewer suggested (DOI: 10.5281/zenodo.10938401). This data will be made publicly available upon the publication of our manuscript.

2. The authors didn't I use CheckM during assembly. Considering the importance of CheckM in assessing the quality of assembled genomes and identifying potential contamination, the authors are encouraged to use it. Additionally, how do the authors ensure the reliability of the assembled genomes and the accuracy of downstream analyses?

We did use CheckM to perform quality assessment on assembled genomes. The assessment evaluated their completeness, contamination, and more. We have presented the CheckM calculation results for all bins in **Supplementary Table 5** in the revision. We have added information on genome assembly in the method section of the revised manuscript (section of Metagenomic binning). Multiple quality control steps are also detailed in the additional method section.

3. What is the justification for using male animals and not a mixed population? Does Cholestasis primarily occur in male population?

Female mice have cyclical fluctuations of estrogen during their estrous cycle, and estrogen levels can influence cholestasis (doi.org/10.1016/j.ajpath.2023.06.010). To avoid the potential impact of cyclical estrogen fluctuations on cholestasis, we used male mice in this study. Further, what we had in mind regarding the human relevance of this study was about patients with non-pregnancy-related cholestasis.

4. Please elaborate on the histopathological methods as it is not clear in the methods part. How was the scoring done and was it a single / double not blinded

scoring or was it performed by a trained mouse pathologist. Also please explain the scoring system to assess tissue damage.

The detail of immunohistochemistry experiment is as follows: We removed the left lateral lobe, the largest hepatic lobe of the mouse liver, from each mouse and the center portion (~ 5 mm x 5 mm x 5 mm) of the lobe was cut and fixed in 4% paraformaldehyde fixation buffer for 48 hours. After dehydration, the specimens were embedded in paraffin blocks and labeled with random serial numbers. The blocks were sectioned by an experienced rodent pathologist in Guge Biotechnology Co., Ltd. The tissue sections were stained, analyzed and scored based on the commonly used Suzuki's score method (**Table R1 below**) by the pathologist who was completely blinded to our experiments. Five representative microscopic fields of each section were photographed using a Leica DMRBE microscope. Image Pro Plus 6.0 was used to calculate the Integrated Optical Density (IOD). The results were sent back to the laboratory. Thus, the scoring was done double blinded whereas the tissue block label was single blinded.

Score	Congestion	Cytoplasmic vacuolization	Parenchymal necrosis
0	No	No	No
1	Minimal	Minimal	Single-cell necrosis
2	Mild	Mild	< 30%
3	Moderate	Moderate	< 60%
4	Severe	Severe	> 60%

Table R1 | Suzuki's score.

5. The authors have an excellent set of data from different related microbial niches, and it would be a missed opportunity not describing each microbiota profile properly. Although, the authors have made commendable efforts on presenting their data but for exploratory purpose of the microbiota, it would be

informative to construct prevalence and abundance plots of each microbial niche at genera / species in Figure 2. As a reader, I would appreciate such descriptive plots and will provide me with the idea on which species are the primary members in gut.

We thank the reviewer's suggestion and have added analyses of the relative abundance of different intestinal microbial taxonomic levels (phylum, class, order, family, genus). We presented those results in the supplementary Figures (Fig. S3I-M).

6. In Result / Figure 2, the authors didn't choose to use the conventional Shannon and Simpson index in the main figure. It is important to show at least the Simpson diversity index should be used. This is a mathematical measure used in ecology to quantify the diversity or richness of species in each community or ecosystem. It provides insights into how evenly individuals are distributed among different species in a community. A higher Simpson diversity index indicates lower diversity because it means that there is a higher probability that two randomly selected individuals will belong to the same species. In this study the Simpson index is higher in NC, which is counter intuitive. In general, there are two ways one can calculate this:

i. Simpson's Index (D) = $1 - \sum(p_i^2)$

where:

- D is the Simpson diversity index.

- p_i is the proportion of individuals belonging to the i-th species.

This index quantifies the dominance of a few dominant species in a community.

ii. Hill Numbers (Diversity Order q):

Hill numbers provide a family of diversity indices that can be tailored to different research questions by varying the diversity order "q."

If the authors are primarily interested in understanding dominance and the influence of a few dominant species in your community, Simpson's index (D) might be more appropriate. If you want to consider both the richness and

evenness of species in your community and have the flexibility to emphasize one aspect over the other, Hill numbers with different "q" values can be a powerful choice.

Please indicate which method was used due to contradictive results.

We thank the reviewer for the professional and meticulous comments. In fact, we had conducted six types of alpha diversity assessments on the gut microbiota structures of the two groups of mice, which are Chao1, Observed Species, Simpson, Goods Coverage, Pielou e, and Shannon Index. Except for Chao1 and Observed Species that were shown in **Fig. 2A & B**, the rest were shown in **Fig. S3** in the original version of the manuscript. We acknowledge the reviewer's comments and will move Simpson to the main figures as **Fig. 2C**.

7. Regarding Result / Figure 2, the authors use Chao1 index. It is arguably a flawed method of indicating species richness. It is sensitive to number of taxa appearing in each sample and sequencing depth achieved. This means that two samples when subjected to the same sequencing effort may produce different richness values. Are the alpha diversity hill indices? Shannon and Simpson indices are more robust and should be used here. Please use the Renyi-Hill number calculation in vegan package for this (Hill numbers 1 to 4). Please note that Shannon diversity in other packages such as phyloseq and QIIME is not the same as Hill indices, which are more comparable between each other. For help follow these articles:

<https://esajournals.onlinelibrary.wiley.com/doi/10.1890/13-0133.1>

<https://www.nature.com/articles/srep38263>

Thank you for your comments. Indeed, we had applied six types of alpha diversity assessments for the gut microbiota structures of the two mouse groups, in responding to the earlier comment. The outcomes were consistent using the different assessments (**Fig. 2A-C** and **Fig. S3A-C**). We now have incorporated the Hill indices method for calculating the alpha diversity between two groups in the revised manuscript (**Fig. 3D-G**). Thus, we have used 10 different methods to calculate the alpha diversity

between the two groups and the results are consistent.

8. The authors chose to use Bray-Curtis dissimilarity for beta diversity, which is a straightforward metric based on the relative abundances of taxa and does not consider phylogenetic relatedness. However, performing UniFrac analysis that considers species phylogeny will provide a much higher resolution data on the contribution to the disease/condition. This becomes more important since the authors observed changes in microbiota diversity with lung adenocarcinoma. UniFrac incorporates phylogenetic information, which means it considers the species and evolutionary relatedness. UniFrac distances are robust to differences in sequencing depth, making them suitable for comparing samples with varying levels of sequencing coverage. UniFrac distances are more biologically meaningful because they reflect the genetic divergence between microorganisms. I would suggest performing this alongside BC-measure and report pairwise PERMANOVA between conditions.

In addition to calculating beta diversity using Bray-Curtis dissimilarity, we added the calculation of beta diversity using Weighted UniFrac based on the reviewer's comments. By calculating beta diversity with Weighted UniFrac distance (**Fig. 2F** and **Fig. 2G**), we found the results to be consistent with those obtained using Bray-Curtis dissimilarity. This indicates that the differences in beta diversity between the NC and ANIT groups are very stable.

9. In result / figure 3, were these groups done in a separate experiment or together with the experiments performed for Figure 1 and 2. If not then did the authors look for similar effects on Day 2 and Day 8 using basic measurements? which would be consistent with the observations with Day 14 and Day 28.

Short-term and long-term experiments were conducted separately. In the long-term experiment, we did evaluate day 2 and day 8 using the same basic measurements including body weight, water intake, and food intake, all of which showed similar results comparing to the mice in short-term experiment at the same time points.

10. In result / figure 3, why did the authors choose to separate Day 2 - Day 8 and Day 14- Day 28 and not put it in the same comprehensive microbiota analysis as Figure 2? Why wasn't TNF α measure in Figure 2?

We do not think that we can merge the data from Day 2-Day 8 with that from Day 14-Day 28 as they were from 2 sets of experiments and represented two distinct stages of the disease. Day 2 and 8 represented the short-term effects on acute liver injury by ANIT (Day 2) and recovery phase of the acute liver injury (Day 8), whereas Day 14 and 28 represents the long-term effects after the recovery of hepatic injury by ANIT. Moreover, the pathophysiological indicators differ in investigating short-term and long-term effects. In the long-term experimental period, the accumulation of bile in the mouse liver has already recovered (**Fig. 3A-C**). To probe any additional inflammatory factors post-acute hepatic injury, we assessed TNF α and found an increased TNF α when the hepatic injury was recovered. Considering the reviewer's comments, we tested the TNF α concentration in the serum of mice during the short-term phase and as expected, the TNF α levels in the ANIT group mice on Day 2 and Day 8 were significantly higher than those in the serum of the NC group mice (**Fig. 1F**). In addition, we measured the TNF α concentration in the serum of germ-free mice that were transplanted with feces from either the NC group or the ANIT group mice. We found that the TNF α levels in the serum of the GF_{ANIT} group mice were also significantly increased (**Fig. 5O**). This indicates that the intestinal microbiota from the donors with cholestasis can cause systemic inflammation in the recipients.

11. In result / figure 3, the authors should elaborate in the methods more on the method used for constructing this genome as it is not clear. Were MAGs constructed for this? Instead of sequence matching ASV136078 the authors should use pipeline like MetaPhlan or Kaiju that will provide community data as well as metagenome data. Then these metagenomes can be used to reconstruct MAGs for E. coli / Shigella. Performing a more metagenome-based phylogeny will also remove the ambiguity of whether it is a Shigella or E. coli. The use of

multiple tools is encouraged. Upon successful MAGs one can now produce phylogenetic trees to locate these bacteria from an evolutionary standpoint.

We appreciate the reviewer's suggestion and have added a "Metagenomic binning" section to the "Methods" in the revision.

In metagenomic data processing, although Metaphlan and Kaiju directly annotate species from reads, our methodology involves initially assembling all reads, followed by species annotation of the resultant metagenome-assembled genomes (MAGs), which identified high-quality bins annotated as *E. coli*/*Shigella*. This approach offers several benefits over direct read annotation: 1) It increases annotation accuracy by assembling reads into longer contiguous sequences or complete genomes, enhancing species identification through more comprehensive sequence information. 2) It increases genome completeness, potentially uncovering low-abundance species overlooked in direct read annotation. 3) It improves functional prediction, as annotations based on more complete genomes are generally more accurate. 4) It reduces species redundancy, avoiding the potential for counting the same species multiple times from reads originated from the same genomic region, thereby minimizing redundancy and bias. We constructed phylogenetic trees using the bins (a total of 9 bins) that could extract 16S rRNA gene sequences, and found that bin 255 had the closest phylogenetic relationship with ASV136078 (**Figure 4B**). Further, sequence alignment revealed that the 16S rRNA gene sequences of bin 255 and ASV136078 were 100% identical (**Figure. 4C**).

12. The authors here indicate that it is the same ASV of *E.coli*-*Shigella* here. Were these sequencing analysis performed together with the ones from previous figure? The authors here choose to perform amplicon sequencing in this case as well. This is of course needed for a global picture. However, after performing metagenomic based identification of *Shigella* / *E. coli* and reconstruction of genomes. The authors should choose a different marker than 16S rRNA and a gene from *E. coli* *Shigella* genome that they have got to precisely identify if it is the same genotype that came up in the previous experiments. This should be used

to perform unique-gene based amplicon sequencing.

All 16S rRNA sequencing data were analyzed together, and having a single ASV number means the sequences are identical. In our amplicon sequencing, we amplified the V3-V4 region of the 16S rRNA gene from the fecal DNA. Although we cannot fully guarantee that any bacteria identified with the same ASV (ASV 136078) would be the same bacterial strain, the V3-V4 region of the 16S rRNA gene includes both variable and conserved regions. When all sequences in the V3-V4 region are consistent, it significantly increases the likelihood that they come from the same bacterial strain. We think that the reviewer's suggestion to choose a marker other than 16S rRNA for the identification would not resolve this issue. The ultimate solution to address this issue would be to isolate and culture the bacterial strain represented by ASV136078 from the fecal samples, followed by whole-genome sequencing of the single bacterium. However, we did try anaerobic isolation of the target bacterial strain for over 6 months without satisfactory outcome. Therefore, based on metagenomic data and using ASV 136078 as a foundation, we found a 16S rRNA sequence of a draft genome (bin255) is 100% match ASV136078. We also analyzed the virulence genes in the draft genome.

13. Why fecal microbiome transfer in short term experiment not in long term? Did the authors perform long term experiments to investigate legacy effects. This effect must be shown to claim long-term effects in the article title.

The acute hepatic injury induced by ANIT is self-limiting and some of the clinical and pathological phenotypes recover on Day 8 as shown in Figure 1. The purpose of the fecal microbiome transfer (FMT) using the Day 2 short-term donor mice was not only to assess the altered gut microbiota, more importantly was to test the effect of altered gut microbiota on the hepatic inflammation, which was not evident in the long-term experimental hosts (**Fig. 3A-C**). Although the change of gut microbiota remained in the long-term experimental hosts, the magnitude of alteration was somewhat attenuated comparing to that in the short-term acute phase. If the reviewer wishes, we will modify the title of our manuscript.

14. It is not clear if pro-inflammatory cytokines were also high upon gut dysbiosis (colonic) measured from fecal samples except for Figure 5. Although the authors show ileal and hepatic expression of inflammatory signals, this lacks for Figure 2, which establishes the foundation of this article. In addition, the random forest models can used to indicate the bacterial species associated with both hepatic and colonic gene expression. This becomes more important since one would expect a case of bacterial diarrhea. I would urge the authors to show this if possible to make a strong case of collateral effects of gut dysbiosis.

We assess the pro-inflammatory cytokines TNF α in the long-term (Day 14 and Day 28) experiment shown in **Fig. 3E**. However, the results from the FMT experiment in germ-free mice, using the Day 2 donor mice in the short-term experiment, lend strong support for the collateral effects of gut dysbiosis. To address the reviewer's concerns, we measured the TNF α concentration in the serum of mice during the short-term phase and in the serum of germ-free mice that were transplanted with the fecal microbiota collected from Day 2 donor mice. We found that the TNF α levels in the ANIT group mice on Day 2 and Day 8 were significantly higher than those in the serum of the NC group mice (**Fig. 1F**). Moreover, the TNF α levels in the serum of the GF_{ANIT} recipient mice were significantly increased (**Fig. 5O**).

Reviewer #3 (Comments for the Author):

Title: Bile Acid-Gut Microbiota Imbalance in Cholestasis and Its Long-term Effect in Mice

In this study, the authors characterized host phenotypes and the microbiome in an induced short-term hepatic cholestasis mouse model induced by α -Naphthyl-isothiocyanate (ANIT). First, the authors did a thoughtful confirmation of acute liver injury in mice by ANIT. They also showed a different microbial composition between mice treated with ANIT and control mice by characterizing the small intestine content and fecal samples using 16S rRNA

amplicon sequencing. Specifically, the authors identified a correlation between *Escherichia-Shigella* and certain bile acids, including secondary bile acids. Then, the authors showed the persistence of various pathologies in non-hepatic organs and tissues over a more extended period in their model, including systemic inflammation by higher levels of circulating TNF α . Regarding the microbiome, the authors showed a persistent bacterial load in fecal and ileal content. Interestingly, germ-free mice colonized with fecal samples for mice treated with ANIT trigger damage and hepatic inflammation. This host response is not observed in germ-free mice colonized with fecal samples for control mice.

This is a relevant study that established a simple mice model system to determine interactions between cholestasis and the gut microbiome compared to other systems, such as bile duct ligation. Using this model, the authors showed an unexpected persistence of pathologies on non-hepatic organs and tissues and dysbiosis over a long period after the self-limiting periods, underlining the relevance of recovery periods to manage the disease. However, I have some concerns about the microbiome data analysis.

We are grateful for the reviewer's positive feedback about our work.

Comment 1.

The authors argue that there is no change in the control group's microbiome structure over time and decided to merge all the control samples for statistical analysis (Lines 340-342). However, looking at the PCo plot in Fig 2C, I disagree with the authors. There is a variability of the control samples over the second coordinate in the PCo. By analyzing the PCo plot, I believe that the ANIT treatment induces the development of a different microbiome state. Still, I do not consider the best strategy to merge all the control data for statistical analyses. Please provide a strong argument for this decision or consider doing all the statistical analyses without merging the control samples.

Considering the reviewer's comments, we reanalyzed the fecal samples from the NC

(normal control) group of mice from Day 0, Day 2, Day 4, and Day 8 (4 time points). Using PERMANOVA testing, we found no significant differences in the gut bacterial community structure in the fecal samples of NC group mice across these time points. This result has been added to the supplementary figures (**Fig. S3H**). As we aimed to identify the gut microbiota differences between the cholestasis mice and the control mice, merging the control samples, which have no significant differences among them, will allow us to pinpoint the bacteria that display statistical differences in the ANIT mice. In that sense, the “noise” of the non-significant variability in the control mice would make the analysis more stringent. Should we were to analyze the control group at each time point separately, it could lead to the incidental results due to the partiality of the microbial structure at a given time point.

In the mice experiments, the authors randomize the selection of mice to create two groups: (i) normal control (NC) group and (ii) ANIT group. This strategy should avoid bias in the starting mice microbiome, such as maintaining mice in different cages before the experiment. According to Fig S3, the authors collected fecal samples at time 0 and processed those samples. It is important to show the microbiome data of time 0, in for example Fig 2A and Fig S3A-D, to confirm that mice for both groups started with a similar microbiome and that the observed microbial dysbiosis in the ANIT group corresponds to the treatment.

Our apology for the confusion. The mice were maintained in different cages before the experiments. The mice were randomly selected into two groups (control and ANIT groups) just before the experimentation to minimize the cage effect and there were no significant differences in the structure of gut microbiota between the two groups of mice on Day 0. We have made the corresponding adjustments to **Fig. 2A-C** and **Fig. S3A-G**.

Comment 2.

There is an unusual pattern on day 4 in the short-term experiments. The alpha diversity values of the control sample (Fig 2A and 2B) on day 4 seem lower than

those on days 2 and 8. In addition, the food intake on day 4 for the control sample is lower than the other days. As a control, I would not expect variability in the samples. Are Chao1 and Observed species on day 4 statistically different from days 2 and 8? If so, do you have any idea of this variability? In the material and method section, it says that this experiment was repeated one time. Is this data a mix of both experiments?

For Chao1, the *P*-value between Day 4 and Day 2 in the NC group was 0.0646, and the *P*-value between Day 4 and Day 8 in the NC group was 0.0813. For Observed species, the *P*-value between Day 4 and Day 2 in the NC group was 0.1861, and the *P*-value between Day 4 and Day 8 in the NC group was 0.1264. There was no statistical difference between Day 4, Day 2, and Day 8.

Yes, we combined the results of the two sets of experiments and the data presented were pooled results. Regarding the reduced food intake, we were unable to identify the cause. The method that we assessed the food intake was also simple - by weighing the daily remained food and the weight of the food provided daily.

Comment 3.

The current Fig S4 is of low quality. Please upload a figure with a better resolution.

We have uploaded the figure with higher quality.

Comments 4.

In Fig 2A and 2B, the authors showed a recovery of the alpha diversity metric at day 8 in the ANIT group. However, the alfa diversity metric decreased again on Day 14 and Day 28. Is this correct? If so, do you have any hypothesis of this dynamics (recover and then collapse of the richness)?

Experimental data from **Fig. 2A** and **Fig. 2B** revealed no significant difference in the alpha diversity of the intestinal microbiota between the two groups of mice on day 8, the time when the acute hepatic injury was already on the mend. There was also no significant difference between the two groups on Day 14 (**Fig. 3H** and **Fig. 3I**).

Interestingly, the alpha diversity of the intestinal microbiota in the ANIT group significantly decreased again on Day 28 (**Fig. 3H** and **Fig. 3I**). We hypothesize that cholestasis affects the structure of the intestinal microbiota (including alpha and beta diversity) in our model. Bile acid in the liver showed no difference between the control and ANIT group from Day 8 (**Fig. 1K** and **Fig. 3A**). However, at this time, the total bile acids in the intestines of the ANIT group remained significantly lower than that in the NC group (**Fig. 1L**). This suggests that cholestasis in the ANIT group mice was compartmentally recovered on Day 8. Even though the alpha diversity of the intestinal microbiota in the ANIT group showed no significant difference between the two groups on Day 8 (**Fig. 2A - C**), the structure of the intestinal microbiota remained significantly different (**Fig. 2D-G**). The disruption of the intestinal microbiota structure caused by cholestasis (especially in terms of beta diversity), small intestinal bacterial overgrowth (SIBO), and the increase in opportunistic pathogens, represented by *Escherichia-Shigella*, have a long-term effect. All of which could collectively affect the homeostasis of gut microbiota in the ANIT treated hosts, hence, the structure of the intestinal microbiota (alpha diversity on Day 28) (**Fig. 3H** and **Fig. 3I**). We added a schematic figure below to illustrate the hypothesis (**Fig. R1**).

Comments 5.

For the germ-free experiments, the authors showed a dissimilarity of microbiomes between the one recovered from germ-free mice inoculated with feces from control mice and the one recovered from germ-free mice inoculated with feces of ANIT mice (Fig 5A-D). However, the authors also need to show that the microbiome engrafted in the corresponding germ-free mice is similar to the original inoculum to confirm whether the engraftment was successful or not.

A similar comment was also raised by the Reviewer 1 (comment #1). We compared the gut microbiota structure between donors and their corresponding recipients. Although the gut microbiota between donors and recipients were not completely overlapped, it is clear that the gut microbiota structure of donor mice from the NC group is more similar to that of recipient mice transplanted with fecal samples from

the NC group, and the gut microbiota structure of the donor mice from the ANIT group is more similar to that of recipient mice transplanted with fecal samples from the ANIT group. We have added this set of comparison as new **Fig. 5D** in the revision. The results support our success of the microbiota engraftment. It is noteworthy that we used the pooled fecal samples from NC and ANIT donors, respectively, to ensure the same NC or ANIT source was transferred to individual germ-free mice. We conducted a comparative analysis of the fecal samples between the donors and the recipients. The results of the Principal Coordinates Analysis (PCoA) scatter plot clearly showed that the microbial community structure of the GF_{NC} group was similar to that of its donor, and similarly, the microbial community structure of the GF_{ANIT} group was similar to that of the donor source (**Fig. 5E**). By quantifying the distance among the points on the PCoA scatter plot, we also found that the gut microbiota structure of the recipient mice was very close to their respective donor's microbiota structure (**Fig. 5F** and **Fig. 5G**). These data indicate that the fecal microbiota transplantation experiment was successful.

We have modified **Fig. 5** and the corresponding figure legends.

Comment 6.

In the discussion section (Line 518-521), the authors argue, "Our observation suggests a critical role for bile acids in reducing the abundance of intestinal pathogens and raise the possibility that cholestasis may be a significant contributor to the increase in opportunistic pathogen abundance." This study highlights a broad physiological effect in the ANIT mice, including non-hepatic diseases and not only changes in bile acids. The authors indeed showed the impact of cholestasis in gut microbiome composition, but there is no evidence of direct role of bile acid in changing the microbiome. I would consider tone-down this sentence.

We acknowledge the reviewer's comment and revised the original sentence to " Our findings indicate that bile acids and cholestasis likely influence the composition of the gut microbiome, which may affect the abundance of intestinal pathogens. This

suggests a possible role for cholestasis in altering microbial dynamics, although direct evidence linking bile acid levels to specific changes in the microbiome is yet to be established. Further research is needed to elucidate the mechanisms involved."

Comment 7.

The important section is missing.

We do not understand which section the reviewer was referring to.

Reviewer #4 (Comments for the Author):

The authors assessed gut microbial dysbiosis and bile acid profiles in a drug-induced cholestasis mouse model. The paper not only analyzed the phenotype and microbiome of ANIT-treated mice but also conducted the analysis of genes encoding microbial virulence factors and tried to establish causality using germ-free mice. These aspects merit commendation. However, a significant concern in this paper is the adoption of an oral administration model for the drug (ANIT) despite focusing on gut microbial dysbiosis in cholestasis. This is problematic as it cannot rule out the direct impact of the drug on the microbiome (indeed, the possibility is quite high) and is not an ideal model for studying the relationship between the disease and dysbiosis. Moreover, they have not clearly stated the hypothesis and/or objectives of this study, making it unclear what the specific research aims are. Below is a list of specific comments from the reviewer.

We appreciate the reviewer's critical comments. Most of medications taken by oral route are metabolized in the liver and a list of drugs can cause cholestasis and liver damage (DOI: 10.1055/s-0034-1375964, DOI: 10.1016/S2468-1253(20)30006-6, DOI: 10.1016/j.cld.2013.07.015, DOI: 10.1002/hep.24229). Orally administration of ANIT is an established animal model for drug-induced intrahepatic cholestasis (DOI: 10.1007/BF00296964, DOI: 10.1016/j.jep.2015.12.033, DOI: 10.1016/j.biopha.2017.02.084). We hypothesized that drug-induced hepatic injury leads to dysbiosis of gut microbiota, which in turn affect the host liver function. Our research objectives are three fold – 1) to systematically (both large and small intestine)

and dynamically assess the changes in the gut microbiota that are affected by cholestasis; 2) to investigate if the gut microbiota will return to normal state along with the recovery of cholestasis; 3) to determine if the dysregulated gut microbiota due to cholestasis could cause cholestasis and hepatic abnormality in a “naïve” host. We agree with the reviewer that we cannot rule out the possible direct effect of ANIT on the composition of gut microbiota, however, the results from our experiment using germ-free mice provide the evidence for the role of gut microbiota in cholestasis and hepatic abnormality. We have emphasized this in the revised manuscript.

1. As mentioned earlier, the oral administration model of the drug (ANIT) used in this manuscript is not suitable for studying the relationship between the disease and dysbiosis because the direct impact of ANIT on the microbiome is considered substantial.

As discussed earlier, oral administration model of ANIT is a well-established mouse model for acute cholestasis and liver injury (DOI: 10.1016/s0015-6264(77)80029-1, DOI: 10.1093/toxsci/kfp020, DOI: 10.1016/j.biopha.2017.02.084). Also as discussed earlier, most of the medications used in the clinical settings are through oral administration, which is convenient and cost-effective. Liver is the major organ to break down the drugs and a list of orally administered drugs in humans have liver toxicity, whereas fewer studies investigated the role of gut microbiota in the drug-induced liver toxicity. We applied the ANIT-induced acute cholestasis and liver injury model to probe the role of gut microbiota in drug-induced liver injury. We agree with the reviewer that we cannot rule out the possible direct impact of ANIT on gut microbiota. Interestingly, a study by Cullen et al showed that oral ANIT did not have direct effect on gut microbiota when comparing cholestasis and liver injury in germ-free rats with conventional rats, both of which had similar bile duct and liver injury (doi.org/10.1177/0192623316662360). This study lends some support to our finding that cholestasis and liver injury are likely to be the cause, at least in part, of the dysbiosis in this model. Moreover, the results from our current experiment using germ-free mice support the role of gut microbiota in cholestasis and hepatic

abnormality. However, the ultimate proof is to inject (subcutaneously) ANIT into the animals. We revised our discussion by commenting the limitation of our current study.

2. The relative abundance of *Escherichia-Shigella* is presented in Fig 2F and Fig 3L. However, the values for each control group differ significantly. Please provide an explanation for this discrepancy.

We believe the difference is due to individual variability of the mice. The mice used in the long-term experiments (Fig.3) were not from the same progeny of the mice used in the short-term experiments (Fig.2). In addition to the effect of maternal gut microbiota on the progeny, mice housing in different cages can influence their gut microbiota. Since mice are coprophagia, mice in the same cage are more likely share similar gut microbiota. As not all the mice used in the study could be housed in the same cage (the cage density policy is 5 mice per cage), this could also contribute to the variability among the mice used in the study. However, the same “stock” of mice were used in each experiment and NC and ANIT groups were studied in parallel.

3. Fig 2L indicates an increase in *Escherichia-Shigella* in the small intestine, but how did other taxa change?

We have presented the changes in other taxa in the form of a heatmap in Fig. S6B. We focused on *Escherichia-Shigella* (ASV 136078) because it is the bacterial species with the highest relative abundance in the small intestine of the ANIT group (accounting for approximately 10%) on Day 2, the peak of the clinical and pathological manifestations.

4. It is surprising that in the germ-free model experiment, administering faeces without the drug had an impact on liver function and inflammatory cytokines. What is the reason for intentionally not administering ANIT? Additionally, details of this experiment are not outlined in the methods; for example, when was the analysis conducted after the faecal transplantation?

The rationale of the fecal microbiota transplantation (FMT) experiment is to determine the pathophysiological function of the altered gut microbiota due to cholestasis in a “clean” host. Administration of ANIT would override the bona fide pathophysiological function of the altered gut microbiota.

We have more information regarding the FMT experiment in the methods section.

5. In the same experiment (Fig 5), they examined the gene expression of muc1, muc2, and muc3 as mucin-related genes. While the importance of muc2 in mucin layer formation in mice is well-known, explain the rationale behind investigating muc1 and muc3.

As the reviewer pointed out that Muc2 is one of the main secretory mucins, has a significant role in the intestine. However, Muc1 and Muc3, as membrane-associated mucins, also play crucial roles in maintaining intestinal barrier integrity. Muc1 is considered to prevent from pathogen adhesion to the epithelial cell surface (doi.org/10.3389/fcimb.2022.856962, [doi:10.1038/mi.2012.98](https://doi.org/10.1038/mi.2012.98)). Muc3 also has the ability to inhibit pathogen adhesion to the epithelial cell surface ([doi: 10.1136/gut.52.6.827](https://doi.org/10.1136/gut.52.6.827)), promoting wound healing ([doi:10.1053/j.gastro.2006.09.006](https://doi.org/10.1053/j.gastro.2006.09.006)), and regulating inflammatory responses. Muc3 can interact with microbes and other environmental factors through its adhesion region, participating in maintaining the integrity of the intestinal mucosal barrier. (doi.org/10.1016/j.bbagen.2014.05.003). The rationale of investigating Muc 1, 2 and 3 was to assess the tissue response of germ-free host more comprehensively to the altered gut microbiota caused by cholestasis.

6. In the same experiment (Fig 5), was the bacterial community in the small intestine affected? For example, was there any evidence of small intestine overgrowth of bacteria?

We quantified the bacteria in the small intestine (ileum) and found no significant difference in the bacterial load between the GF_{NC} and GF_{ANIT} groups of mice (see the figures below - **Fig. R2A**). We speculated that the prerequisite for small intestinal

bacterial overgrowth could be due to a reduction in intestinal bile acid concentration. However, we did not find significant difference in the intestinal bile acid concentrations between the GF_{NC} and GF_{ANIT} groups of mice (shown in the figure below - **Fig. R2B**).

Fig. R2 | (A) The ileal bacterial content of GF_{NC} and GF_{ANIT} mice as measured by real-time qPCR, n=5/group. (B) Total bile acids in the fecal samples, n=5/group. The data in (A) and (B) are presented as mean ± s.e.m., and Student's t-test (two-tailed) was used to analyze differences between the groups: GF_{NC} vs. GF_{ANIT}.

Re: mSystems00127-24R1 (Bile Acid-Gut Microbiota Imbalance in Cholestasis and Its Long-term Effect in Mice)

Dear Dr. Li Wen:

Your manuscript has been accepted, and I am forwarding it to the ASM production staff for publication. Your paper will first be checked to make sure all elements meet the technical requirements. ASM staff will contact you if anything needs to be revised before copyediting and production can begin. Otherwise, you will be notified when your proofs are ready to be viewed.

It is highly important that you include an Importance section in your manuscript, as it is a requirement for publication in our journal.

Sincerely,
Daniel Garrido
Editor
mSystems

Reviewer #1 (Comments for the Author):

None

Reviewer #2 (Comments for the Author):

The authors have successfully met all the concerns. In addition, the authors have also justified the use of male mice in the animal experiments. I would suggest if the authors haven't already done it, to add this justification as text in the appropriate text in the manuscript. The authors can decide whether to state it in the results or in the methods.

Reviewer #3 (Comments for the Author):

The authors successfully addressed all my concerns, and the additional data/analyses strengthened their manuscript. However, the manuscript lacks the "Importance section" in the two-part abstract. On the mSystems Formatting Information website (<https://journals.asm.org/journal/msystems/article-types>), the authors can check the requirements of the two-part abstracts and the guidance of the Importance section.

Reviewer #4 (Comments for the Author):

Thank you for all the responses to our comments. I agree with your statement in the discussion, acknowledging the limitation of the study, including the fact that we cannot completely exclude the direct impact of orally administered ANIT on the gut microbiota.